# Engineered cytosine base editor enabling broad-scope and high-fidelity gene editing in *Streptomyces*

Jian Wang [1], Ke Wang[1], Zhe Deng[1], Zhiyu Zhong[1], Guo Sun [1], Qing Mei[1], Fuling Zhou [1], Zixin Deng[1] & Yuhui Sun [1,2] ✉

Base editing (BE) faces protospacer adjacent motif (PAM) constraints and off-target effects in both eukaryotes and prokaryotes. For *Streptomyces*, renowned as one of the most prolific bacterial producers of antibiotics, the challenges are more pronounced due to its diverse genomic content and high GC content. Here, we develop a base editor named eSCBE3-NG-Hypa, tailored with both high efficiency and -fidelity for *Streptomyces*. Of note, eSCBE3-NG-Hypa recognizes NG PAM and exhibits high activity at challenging sites with high GC content or GC motifs, while displaying minimal off-target effects. To illustrate its practicability, we employ eSCBE3-NG-Hypa to achieve precise key amino acid conversion of the dehydratase (DH) domains within the modular poly-ketide synthase (PKS) responsible for the insecticide avermectins biosynthesis, achieving domains inactivation. The resulting DH-inactivated mutants, while ceasing avermectins production, produce a high yield of oligomycin, indicating competitive relationships among multiple biosynthetic gene clusters (BGCs) in *Streptomyces avermitilis*. Leveraging this insight, we use eSCBE3-NG-Hypa to introduce premature stop codons into competitor gene cluster of *ave* in an industrial *S. avermitilis*, with the mutant Δolm exhibiting the highest 4.45-fold increase in avermectin B1a compared to the control. This work provides a potent tool for modifying biosynthetic pathways and advancing metabolic engineering in *Streptomyces*.

*Streptomyces* and related genera of Actinobacteria are Gram-positive filamentous bacteria that have produced more than two-thirds of the bioactive secondary metabolites with significant medical and agricultural implications[1,2]. As the preeminent producers of natural products among bacteria, they continue to captivate the scientific community. To harness and manipulate the myriad gene clusters that govern biosynthetic pathways, there is an exigent demand for developing convenient and efficient genome editing tools[3]. Traditionally, for gene editing in *Streptomyces*, single-crossover integration can be used for gene disruption, while double-crossover integration based on homologous recombination is employed in gene knock-out or knock-

in events[4,5]. However, conventional methods are time-consuming, inefficient, and incapable of multiple gene editing.

Recently, the clustered regularly interspaced short palindromic repeat (CRISPR)/CRISPR-associated protein (Cas) system demonstrates its efficacy across a multitude of species, including *Streptomyces*[6-11]. It can expedite genetic manipulation in *Streptomyces* by introducing a DNA double-strand break (DSB) at the target site guided by single guide RNA (sgRNA), subsequently repaired through non-homologous end joining (NHEJ) or homologous recombination, yielding either stochastic insertions and deletions (indels) or precise genome editing proximal to the DSB. However, the effective in situ

[1]Department of Hematology, Zhongnan Hospital of Wuhan University, School of Pharmaceutical Sciences, Wuhan University, Wuhan 430071, China. [2]School of Pharmacy, Huazhong University of Science and Technology, Wuhan 430030, China. ✉e-mail: yhsun@whu.edu.cn

editing capability depends on the efficiency of the intrinsic or reconstituted repair machinery. In contrast to eukaryotes, this process is notably less efficient in bacteria[12] and may exhibit variability among distinct species, thus impeding the application of this system.

More recently, the emergence of base editing (BE), a gene editing technology derived from CRISPR/Cas system, demonstrates notable application potential and significance[13,14]. The fusion of a cytidine deaminase with the Cas9 gives rise to cytosine base editors (CBEs), affording the programmable conversions of cytidine-to-thymidine (C-to-T) mutations within a defined genomic locus without inducing DSBs or template donor DNA[15] (Fig. 1a). Notably, CBEs can markedly enhance editing efficacy in comparison to the conventional Cas9 endonuclease-mediated homology-directed repair approach for the installation of point mutations[16]. In addition, a series of expansions and improvements on the methodology are performed by multiple studies[17–22], and CBEs have been applied in a wide variety of animals[23–25], plants[26–28], and bacteria[29–34]. To date, two classes of CBEs are implemented in *Streptomyces*: the third-generation base editor (BE3), comprising the rat APOBEC1 (rAPOBEC1) cytidine deaminase, nCas9(D10A), and a uracil glycosylase inhibitor (UGI)[35]; and dCas9-CDA-UL$_{str}$ based on Target-AID, which encompasses the nuclease-deficient Cas9 (dCas9), the cytidine deaminase sourced from *Petromyzon marinus* (PmCDA1), the UGI, and the protein degradation tag (LVA tag)[33,36]. Both of these two CBEs are capable of efficient base editing in *Streptomyces*. However, the same component, Cas9 from *Streptococcus pyogenes* (SpCas9), within the two CBEs, strictly recognizes DNA target sites bearing the NGG protospacer adjacent motif (PAM)[37], which occurs at an average frequency of only once in every 16 randomly selected genomic loci[38]. This limitation results in a substantial proportion of desired target sites being rendered inaccessible, thereby impeding the utilization of CBEs. A potential avenue to circumvent this constraint lies in using engineered Cas9 variants with modified PAM specificities. Notable examples include xCas9, SpCas9-NG, and SpRY[38–40]. Specifically, xCas9 and SpCas9-NG exhibit recognition of NGN PAM, whereas SpRY targets NRN (where R represents either A or G) and NYN (where Y signifies either C or T) PAM.

In addition, there is a noteworthy concern regarding off-target effects at both genomic and transcriptomic levels induced by CBEs, including BE3 and Target-AID[41–44]. Several high-fidelity variants of SpCas9 hold the potential to address this problem, including eSpCas9(1.1)[45], SpCas9-HF1[46], HypaCas9[47], evoCas9[48], and Sniper-Cas9[49]. Moreover, the efficiency of BEs is typically affected by various factors, including different deaminases, CRISPR proteins, PAM constraints, editing window-imposed restrictions, DNA motifs, and more[15]. In the case of *Streptomyces*, with a hallmark feature of GC-rich genomes (>70%), DNA motifs may present a bottleneck for the application of BE3. Widely used rAPOBEC1 of BE3 imposes TC motif preferences while exhibiting poor processing of cytosines within GC motifs[13,19], affecting the application of BE3 in *Streptomyces*. Meanwhile, whether the high GC content of protospacers will affect the efficiency of base editing warrants further investigation.

In this study, to expand the targeting range of CBE while maintaining high efficiency and fidelity in *Streptomyces*, we seek to amalgamate the capabilities of the high-fidelity Cas9 variants (SpCas9-HF1 or HypaCas9) and a Cas9 mutant proficient in recognizing NG PAM sequences (SpCas9-NG) into the BE3 platform. We engineer and characterize 11 variants of CBE3 to identify the most suitable base editing tool for applications in *Streptomyces*. We find that high GC content (85%) of protospacer (20 nucleotides) can erode the editing efficiency of rAPOBEC1 based CBE3 system, even down to a negligible value. However, the engineered tool, eSCBE3-NG-Hypa demonstrates high-efficient base editing up to 97.67% at high-activity sites featuring the NG PAM. Meanwhile, it exhibits robust editing efficiency of up to 82.38% at sites with a high GC content of 85%. In contrast to widely used CBE3, eSCBE3-NG-Hypa exhibits high activity at GC motif and

lower bystander effects in the editing window spanning from position 5 to 9 (the PAM-distal position in the protospacer is considered as 1). In contrast to CBE3, the utilization of eSCBE3-NG-Hypa in *Streptomyces* results in an approximately 3.7-fold expansion in optional protospacers. Furthermore, the proportion of genes allowing the introduction of a premature stop codon can be elevated from 92.03% to 98.72% in the *Streptomyces* genome. Moreover, the safety profile of eSCBE3-NG-Hypa is validated through comprehensive whole-genome sequencing (WGS) and RNA sequencing (RNA-seq). Last, we employ eSCBE3-NG-Hypa to modify the polyketide synthase (PKS) and metabolic rewiring to obtain potential derivatives of natural products and further improve avermectins production in the over-producing industrial strain. We successfully increase avermectin B1a, a widely-utilized anthelmintic and insecticidal main compound, to a titer of about 1 g/L in 500 mL shake flask cultivation, representing a 4.45-fold increase compared to the control strain. Our results show that eSCBE3-NG-Hypa induces C-to-T conversions with high efficiency and fidelity at target sites bearing NGN PAM in *Streptomyces*, underscoring its significant potential for biosynthetic pathway modification and metabolic engineering applications.

## Results

### Design of CBEs with expanded targeting range and high-fidelity

We employed a shuttle vector[50] capable of mediating conjugal transfer from *Escherichia coli* into *Streptomyces* to introduce the BE system (Supplementary Fig. 1a). CBE3 was originally developed for use in eukaryotic systems[13]. However, to ensure optimal gene expression and compatibility with *Streptomyces*, CBE3 was subjected to codon optimization for this bacterial species. This optimization process resulted in the creation of *Streptomyces* CBE3 (SCBE3) (Fig. 1b). Given the relatively lower efficiency exhibited by xCas9 and concerns regarding the higher off-targeting effects associated with SpRY[38], we opted for the utilization of SpCas9-NG to confer the ability to recognize NG PAM to SCBE3, leading to the development of SCBE3-NG (Fig. 1b). To develop CBEs with high-fidelity, SpCas9-HF1, and HypaCas9 were selected due to their well-balanced overall activity and specificities[51]. Consequently, we obtained SCBE3-HF1, SCBE3-Hypa, and SCBE3-HF1-Hypa (Fig. 1b). In pursuit of CBEs featured by both high-fidelity and an expansive recognition scope, we incorporated substitutions of SpCas9-NG into the SpCas9 portion of SCBE3-HF1, SCBE3-Hypa, and SCBE3-HF1-Hypa, thus constructed SCBE3-NG-HF1, SCBE3-NG-Hypa, and SCBE3-NG-HF1-Hypa (Fig. 1b). All fusion proteins and sgRNA cassettes are driven by the constitutive promoter *rpsL*p(XC)[52] and *kasO*p*[53], respectively (Fig. 1b). For sgRNA cloning, we implemented a multiple sgRNA cloning cassette (MSCC) which adapts the 20-nt spacers as homologous overlaps for Gibson assembly, thereby facilitating the straightforward cloning of either single or multiple sgRNAs (Supplementary Fig. 1b–d).

### Broad targeting scope of SCBE3-NGs in *Streptomyces coelicolor*

In our purpose to thoroughly investigate the editing efficiency of CBE variants across target sites featuring diverse PAMs and varying GC contents, we selected 16 protospacers characterized by CGA, CGT, CGC, and CGG PAMs, along with gradient-increasing GC content of 55%, 65%, 75%, and 85% in nonessential gene of the *Streptomyces* model strain *S. coelicolor* M145 (Supplementary Data 1). Given that the editing efficiency of CBE at different protospacers is influenced by position of the cytidine within the editing window and motif differences[13], we ensured that all 16 selected protospacers shared identical sequence characteristics: NNDDTCDDDNNNNNNNNNNNCGN (where N is A, T, C, or G, and D is A, T, or G), with a fixed position for the editable cytidine and consistent TC motif.

For a comprehensive evaluation of editing efficiency across a substantial population, genomic DNA was extracted from a composite of over 100 exconjugants. Next-generation sequencing (NGS) analysis

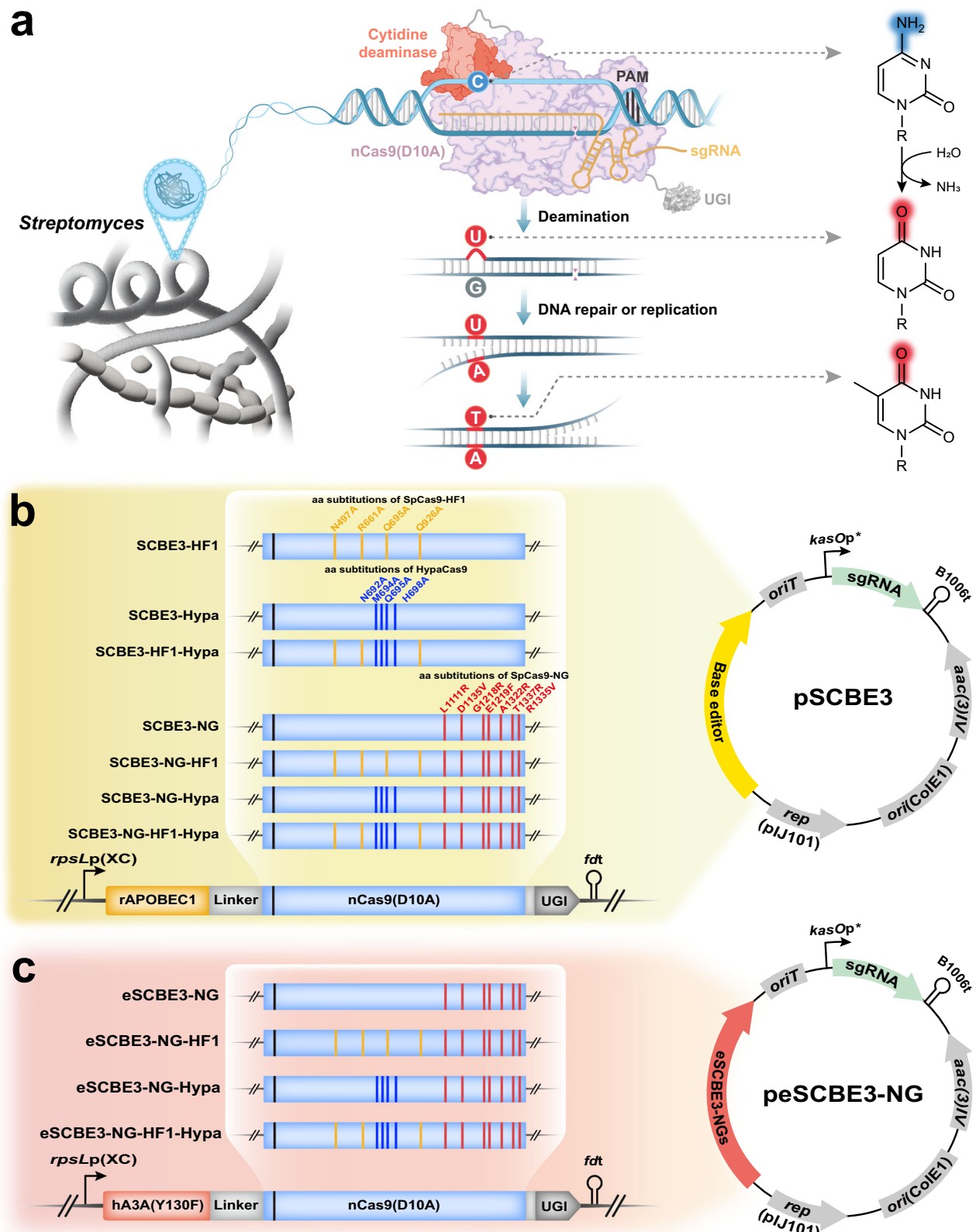

**Fig. 1 | Representation of the base editing strategy and engineering of the *Streptomyces* base editing system. a** Mechanism of CBE. The nCas9(D10A):sgRNA:DNA complex provides single-strand DNA substrate for the deaminase. The chemistry of the base conversions is illustrated on the right side. nCas9(D10A) nicks the target strand to make U•G mismatch conversion into U•A mismatch conversion more effective. UGI protects the generated U bases from being excised by endogenous uracil DNA glucosylase. **b** Schematic of the engineered base editors based on nCas9(D10A) of SCBE3 system. **c** Schematic of the engineered base editors based on SCBE3-NG. hA3A(Y130F), the cytidine deaminase human APOBEC3A(Y130F). nCas9(D10A), Cas9 nickase with D10A mutation; PAM, protospacer adjacent motif; UGI, uracil glycosylase inhibitor; *rpsL*p(XC), the *rpsL* promoter from *Xylanimonas cellulosilytica*; *fd*t, the fd terminator; *kasO*p\*, *Streptomyces* constitutive promotor; B1006t, B1006 terminator; rAPOBEC1, the cytidine deaminase rat APOBEC1; The portion corresponding to nCas9(D10A) of the base editors in **b** and **c** is depicted as blue rectangles, wherein the vertical black lines indicate the D10A mutation site, the vertical red lines signify the mutation sites R1335V/L1111R/D1135V/G1218R/E1219F/A1322R/T1337R of SpCas9-NG, the vertical yellow lines denote the mutation sites N497A/R661A/Q695A/Q926A of SpCas9-HF1, and the vertical blue lines represent the mutation sites N692A/M694A/Q695A/H698A of HypaCas9. Plasmids encoding the base editors are illustrated on the right side and in Supplementary Fig. 1a.

revealed that SCBE3-NG, SCBE3-NG-HF1, SCBE3-NG-Hypa, and SCBE3-NG-HF1-Hypa showed comparable C-to-T conversion rates to SCBE3, SCBE3-HF1, SCBE3-Hypa, and SCBE3-HF1-Hypa at four sites bearing NGG PAM (Fig. 2a). However, only SCBE3-NGs (refer to SCBE3-NG, SCBE3-NG-HF1, SCBE3-NG-Hypa, and SCBE3-NG-HF1-Hypa) demonstrated proficiency in modifying sites with NGA, NGT, and NGC PAMs. In contrast, SCBE3, SCBE3-HF1, SCBE3-Hypa, and SCBE3-HF1-Hypa exhibited inefficient editing efficiency at NGA sites and no activity at NGT or NGC sites (Fig. 2a). Additionally, consistent with prior reports[40], SCBE3-NGs exhibited a preference for NGT and NGG sites, albeit with reduced efficiency at NGA and NGC sites (Supplementary Fig. 3a). Furthermore, it was observed that the fidelity-enhancing substitutions of SpCas9-HF1 and HypaCas9 can erode the editing efficiency of SCBE3-NG-HF1, SCBE3-NG-Hypa, and SCBE3-NG-HF1-Hypa (Fig. 2a). Across the four sites with 85% GC content and bearing NGA, NGT, NGC or NGG PAMs, all eight CBE3 variants based on rAPOBEC1 showed low C-to-T editing efficiency (Fig. 2a and Supplementary Fig. 3b), suggesting that the elevated GC content of protospacer hinders the activity of CBEs. Overall, SCBE3-NGs can efficiently edit most sites with NGN PAMs, albeit with a reduced relative efficiency at sites with NGA or NGC PAM. Additionally, they displayed distinctly lower capability on sites with 85% GC content (Supplementary Fig. 3b).

### Enhanced C-to-T editing activities by eSCBE3-NGs compared to SCBE3-NGs

Having established the feasibility of SCBE3-NGs to expand the scope of base editing widely, we anticipated a further enhancement in the overall editing efficiency of SCBE3-NGs, particularly at sites with NGA PAM, NGC PAM, or high GC content. To achieve this objective, we replaced rAPOBEC1 within the SCBE3-NGs with hAPOBEC3A(Y130F), a deaminase with high activity in regions marked by high methylation levels and GC motifs, coupled with low RNA off-target activity[41,54], resulting in the constructions of engineered SCBE3-NGs which refer to it hereafter as eSCBE3-NGs (refer to eSCBE3-NG, eSCBE3-NG-HF1, eSCBE3-NG-Hypa and eSCBE3-NG-HF1-Hypa) (Fig. 1c).

Evaluation of eSCBE3-NGs at 16 sites with NGN PAMs revealed an overall improvement in editing efficiency compared to SCBE3-NGs (Fig. 2a and Supplementary Fig. 2). The mean C-to-T editing rates were observed to be 59.7%, 72.9%, 59.6%, and 77.4% at sites grouped by NGA, NGT, NGC, and NGG PAM, respectively (Fig. 2b). In contrast to SCBE3-NGs, they exhibited mean C-to-T editing rates of 29.3%, 50.6%, 25.9%, and 53.2% at same grouped sites (Fig. 2b). Particularly, at sites with NGA and NGC PAM, eSCBE3-NG, eSCBE3-NG-HF1, eSCBE3-NG-Hypa, and eSCBE3-NG-HF1-Hypa showed mean C-to-T editing rates of 67.3%, 65.4%, 60.0%, and 46.5% at grouped sites bearing NGA PAM, and 77.6%, 49.9%, 66.7%, and 44.1% at grouped sites bearing NGC PAM, respectively (Supplementary Fig. 3a). In comparison, SCBE3-NG, SCBE3-NG-HF1, SCBE3-NG-Hypa, and SCBE3-NG-HF1-Hypa exhibited lower mean C-to-T editing rates of 40.6%, 34.6%, 23.3%, and 18.8% at grouped sites bearing NGA PAM, and 46.8%, 19.4%, 24.7%, and 12.7% at grouped sites bearing NGC PAM, respectively (Supplementary Fig. 3a).

For grouped sites with 85% GC content, SCBE3-NGs exhibited mean C-to-T editing rates of 15.9%, while eSCBE3-NGs demonstrated robust levels of editing, with mean C-to-T editing rates of 64.8% (Fig. 2c). SCBE3-NG, SCBE3-NG-HF1, SCBE3-NG-Hypa, and SCBE3-NG-HF1-Hypa showed mean C-to-T editing rates of 24.9%, 16.0%, 14.4%, and 8.3% at grouped sites with 85% GC content. In comparison, eSCBE3-NG, eSCBE3-NG-HF1, eSCBE3-NG-Hypa, and eSCBE3-NG-HF1-Hypa displayed significantly improved mean C-to-T editing rates of 77.3%, 66.1%, 67.6%, and 48.3%, respectively (Supplementary Fig. 3b).

In summary, eSCBE3-NGs demonstrated superior base editing performance, particularly at sites with NGA or NGC PAM and GC content above 75%. Given the limited activity of SCBE3-NGs, hindering its application in *Streptomyces*, especially when addressing challenging sites or performing simultaneous editing of multiple targets, eSCBE3-NGs emerge as a more favorable option due to their robust catalytic ability.

### eSCBE3-NGs hold application advantages over BE3 in *Streptomyces*

The elevated GC content (>70%) is a distinctive characteristic of *Streptomyces*, potentially affording a greater abundance of NGG PAM or NGN PAM for SpCas9 or SpCas9-NG within the genome compared to other species. To assess the targeting scope of SCBE3 and eSCBE3-NGs, we conducted bioinformatics analysis on the model strain *S. coelicolor* M145. Given that the introduction of premature stop codons through the CBE system is a prevalent method for gene inactivation[35,36], we used the number of genes and protospacers permitting the introduction of premature stop codons as evaluation criteria.

In essence, CAG, CGA, or CAA in the coding strand, encoding the amino acids Gln, Arg, and Gln, respectively, can be edited to stop codons (TAG, TGA, and TAA) via CBE. Additionally, CCA in the non-coding strand can be edited to TCA, CTA, or TTA, leading to the conversion of TGG in the coding strand to the stop codons TGA, TAG, or TAA. A comprehensive analysis of *S. coelicolor* M145, involving the scanning of up to 8152 open reading frames, revealed that 92.03% of these could be subjected to introducing a premature stop codon using SpCas9 in SCBE3. In contrast, the utilization of SpCas9-NG in eSCBE3-NG-Hypa extended this capability to 98.72% (Fig. 3a). Meanwhile, we observed a significant reduction in the number of editable genes as the count of unique protospacers per gene increased for SpCas9, particularly evident in the scarcity of genes containing more than ten unique protospacers per gene (Fig. 3c). Furthermore, the accumulation of genes allowing stop codon introduction, at each defined number of unique protospacers per gene, was notably lower for SpCas9 than for SpCas9-NG (Fig. 3d). This distinction is particularly evident when genes contain at least ten protospacers, showing a substantial difference of up to 4767, calculated as 5962 genes for SpCas9-NG and 1195 genes for SpCas9 (Fig. 3d and Supplementary Data 2). These findings indicate that the application of SpCas9-NG-based base editors could enable more protospacers to fit diverse applications, offering a flexible design framework for the selection of sgRNA with high editing efficiency and minimal off-target effects.

Although SpCas9-NG presents a significantly higher total of accumulated protospacers (162,044) compared to SpCas9 (44,066),

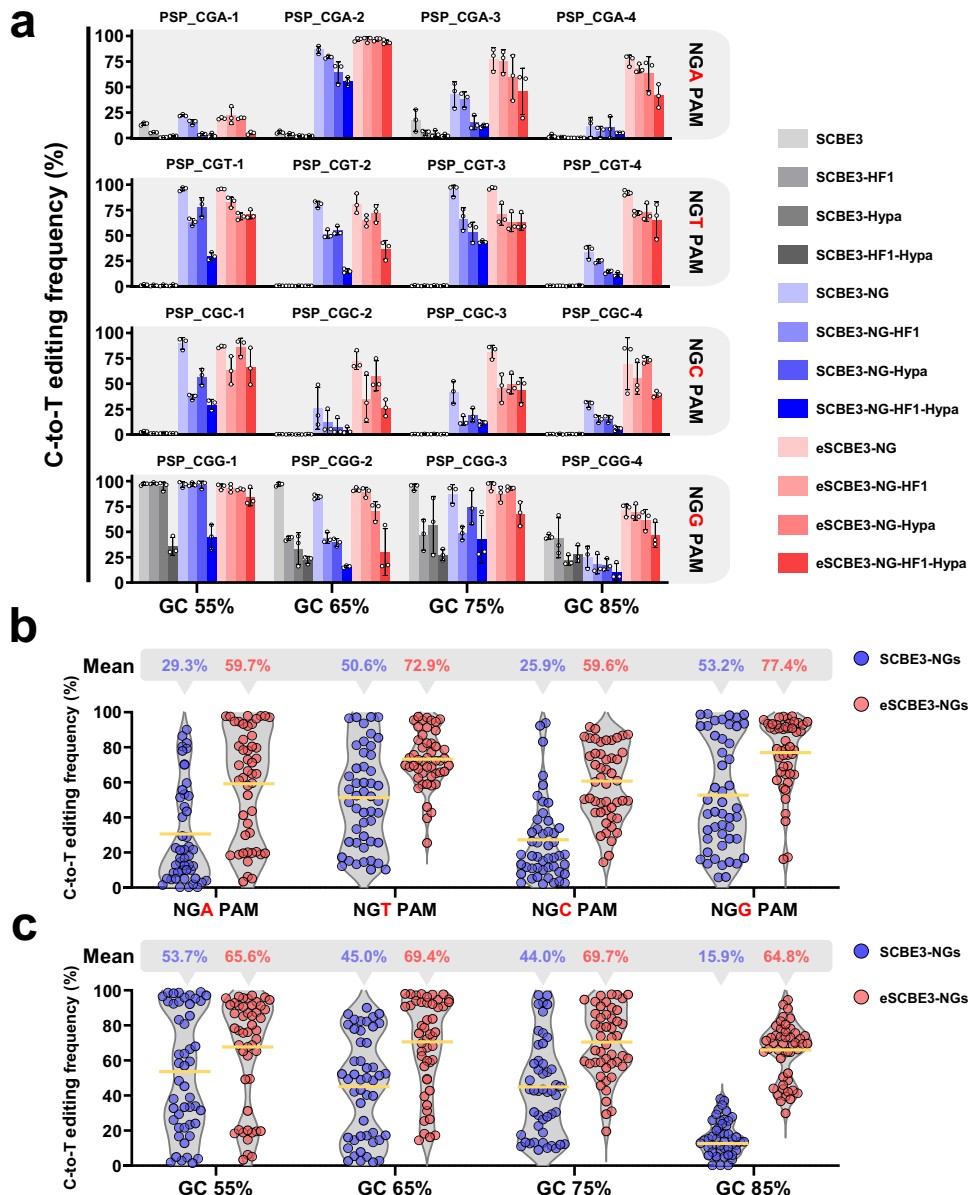

**Fig. 2 | Characterization of CBEs capable of targeting NGN PAMs. a** Modification of 16 protospacers in *S. coelicolor* M145 with NGN PAMs and different GC contents varying from 55% to 85% by 12 CBEs. Bar figures show means and error bars representing s.d. of *n* = 3 independent biological replicates. **b** Summary of the results in (**a**) for SCBE3-NGs and eSCBE3-NGs, but grouped by the NGA, NGT, NGC, and NGG PAM of protospacers. **c** Summary of the results in (**a**) for SCBE3-NGs and eSCBE3-NGs, but grouped by the 55%, 65%, 75%, and 85% GC content of protospacers. **b**, **c** Mean modification (*n* = 48) of SCBE-NGs or eSCBE3-NGs is shown as a horizontal yellow line, and the grey outline is a violin plot.

the distribution of protospacers without a GC motif in the editing window yet with GC content ≤75% is relatively similar, standing at 31.57% for SpCas9 and 32.56% for SpCas9-NG (Fig. 3b). Due to the GC-rich genome of *Streptomyces*, a substantial portion of protospacers features a GC motif in the editing window spanning from position 4 to 8 with the PAM-distal position in the protospacer considered as 1, constituting up to 63.36% for SpCas9 and 62.16% for SpCas9-NG (Fig. 3b). Furthermore, the proportion of protospacers with GC content >75% is 24.28% for SpCas9 and 23.53% for SpCas9-NG (Fig. 3b). In fact, achieving effective C-to-T conversion at a GC motif of protospacers[13,19] or protospacers with GC content >75% has proven challenging for the CBE3 system, let alone for protospacers with both a GC motif and GC content >75%. Whereas, the robust activity exhibited by eSCBE3-NGs makes them more advantageous for applications in *Streptomyces*.

## DNA and RNA off-target evaluation of eSCBE3-NG-HF1 and eSCBE3-NG-Hypa

An essential concern in genome editing lies in the capacity to mitigate potential off-target effects. As eSCBE3-NG-HF1-Hypa exhibits reduced editing efficiency at all examined sites, making it unsuitable for modifying challenging sites and performing simultaneous editing of multiple targets, we selected eSCBE3-NG-HF1 and eSCBE3-NG-Hypa as candidates for further investigation. To comprehensively evaluate potential off-target effects induced by eSCBE3-NG-HF1 and eSCBE3-NG-Hypa, the respective plasmids encoding the base editor and sgRNA targeting the on-target site PSP_CGG-4 were introduced into *S. coelicolor* M145 via conjugation. Subsequently, genomic DNA was extracted from a population of more than 100 exconjugants, and whole-genome re-sequencing was conducted to comprehensively gather information on potential off-target sites in the genome (Fig. 4a). DNA single

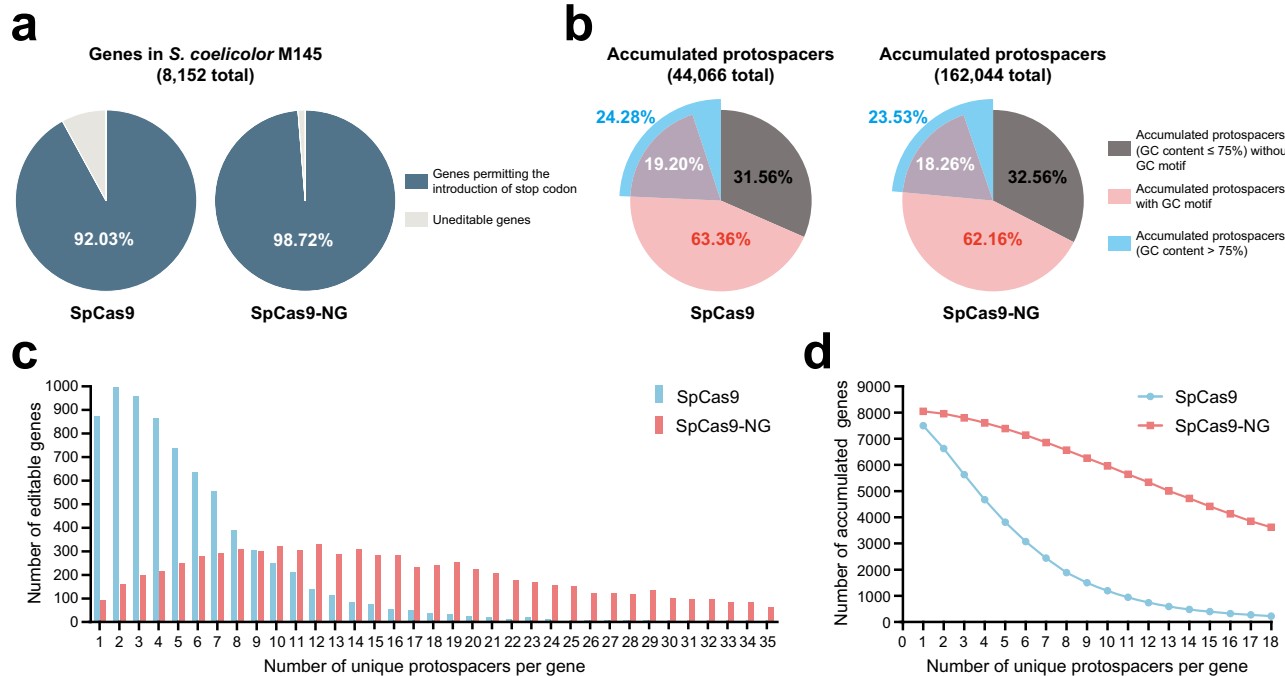

**Fig. 3 | Bioinformatics analysis of eSCBE3-NGs in *S. coelicolor* M145. a** A representation of the percentage of genes permitting the introduction of a premature stop codon in *S. coelicolor* M145 via SpCas9- or SpCas9-NG-mediated CBE is shown. **b** A representation of the distribution of accumulated protospacers enabling the introduction of a premature stop codon from genes in the blue part in (**a**) via SpCas9- or SpCas9-NG-mediated CBE is shown. **c** The number of editable genes that contain a specific number of unique protospacers. **d** The number of accumulated genes that contain more than a specific number of unique protospacers.

nucleotide variations (SNVs) were called from the whole-genome resequencing datasets for each replicate, with the exclusion of any SNVs detected in non-treated samples. In the genomes treated by SCBE3, we found 13. $7 \pm 0.9$ (mean ± s.e.m.) C-to-T/G-to-A conversions, whereas eSCBE3-NG generated a significantly increased number, with $76.7 \pm 9$ (mean ± s.e.m.) (Fig. 4c and Supplementary Fig. 4a). This is ascribed to the higher catalytic activity of hAPOBEC3A(Y130F) and the PAM-relaxed capacity of eSCBE3-NG. However, eSCBE3-NG-HF1 and eSCBE3-NG-Hypa, which combine the functionalities of high-fidelity SpCas9-HF1 or HypaCas9, reduced this number to $49.3 \pm 14.4$ (mean ± s.e.m.) and $26.3 \pm 4.8$ (mean ± s.e.m.), respectively (Fig. 4c and Supplementary Fig. 4a). Furthermore, we observed that eSCBE3-NG led to the high on-target efficiency, while the 87 off-target sites were detected with mean C-to-T/G-to-A editing rates ranging from 4.0% to 92.3% at the genomic level (Fig. 4b and Supplementary Data 3). In comparison, eSCBE3-NG-HF1 and eSCBE3-NG-Hypa displayed comparable on-target efficiency, yet they resulted in 56 and 33 C-to-T/G-to-A conversions, respectively, shared among the 87 identified off-target sites induced by eSCBE3-NG, with mostly much lower frequencies (Fig. 4b and Supplementary Data 3). Regarding the total number of all types of DNA SNVs, eSCBE3-NG-Hypa did not induce significantly more SNVs than SCBE3, but exhibited the lowest number compared to eSCBE3-NG and eSCBE3-NG-HF1 (Fig. 4d).

In addition, we characterized the transcriptome-wide off-target effect of eSCBE3-NG-HF1 and eSCBE3-NG-Hypa. We introduced plasmids encoding SCBE3, eSCBE3-NG, eSCBE3-NG-HF1, or eSCBE3-NG-Hypa, and sgRNA targeting the on-target site PSP_CGG-3 into *S. coelicolor* M145, subsequently harvesting a population exceeding 100 exconjugants for RNA-seq analysis. The results showed eSCBE3-NG-HF1 and eSCBE3-NG-Hypa did not induce significant RNA SNVs compared to the non-treated control (Fig. 4e, f and Supplementary Fig. 4b). However, C-to-U RNA editing was substantially increased in SCBE3-treated genomes, in line with prior reports[41,55], as rAPOBEC1 is known to induce RNA C-to-U editing[56] (Fig. 4e and Supplementary Fig. 4b).

These results indicate that eSCBE3-NG-HF1 and eSCBE3-NG-Hypa induced highly efficient C-to-T base editing at targeted sites, exhibiting markedly reduced off-target effects on a genome-wide scale, along with a background level of RNA off-target mutations. Based on these results, eSCBE3-NG-Hypa was chosen for additional characterization and applications due to its comparable on-target activity but superior performance in reducing off-target effects on a genome-wide scale compared to eSCBE3-NG-HF1 (Fig. 4b–d).

**Characterization of base editing specificity of eSCBE3-NG-Hypa**

To comprehensively assess the base editing capabilities of eSCBE3-NG-Hypa, we selected ten protospacers to evaluate its editing window, motif preference, and bystander activity (Supplementary Data 1). The average editing frequencies of Cs at each position across all ten sites were compared. We observed that eSCBE3-NG-Hypa showed lower editing efficiencies at position 4 and positions 10 to 15, with the PAM-distal position in the protospacer considered as 1 (Fig. 5a). Comparatively, eSCBE3-NG-Hypa displayed higher editing efficiencies spanning from positions 5 to 9, defining the editing window of eSCBE3-NG-Hypa (Fig. 5a).

To investigate the specificity of eSCBE3-NG-Hypa, NGS analysis revealed its selective editing of C at position 9 in the PSP_EW1 site, C at position 7 in the PSP_EW4 site, and C at position 8 in the PSP_EW9 site with significantly high efficiency (Fig. 5b). In contrast, SCBE3 non-selectively edited consecutive Cs at positions 5, 6, and 7 in the PSP_EW1 site and three Cs at positions 5, 6, and 8 in the PSP_EW9 site (Fig. 5b). In certain instances, bystander editing, the additional conversion of bases other than the target base, can occur when multiple editable Cs are present within or near the editing window. Minimizing bystander editing is crucial in specific applications, such as disrupting promoters or regulatory sequences or inducing gene function knockout via the introduction of premature stop codons. At PSP_EW1 and PSP_EW4 sites, both containing the cognate C at position 7 of the protospacers, the cognate-to-bystander editing ratios (C7: C6) induced

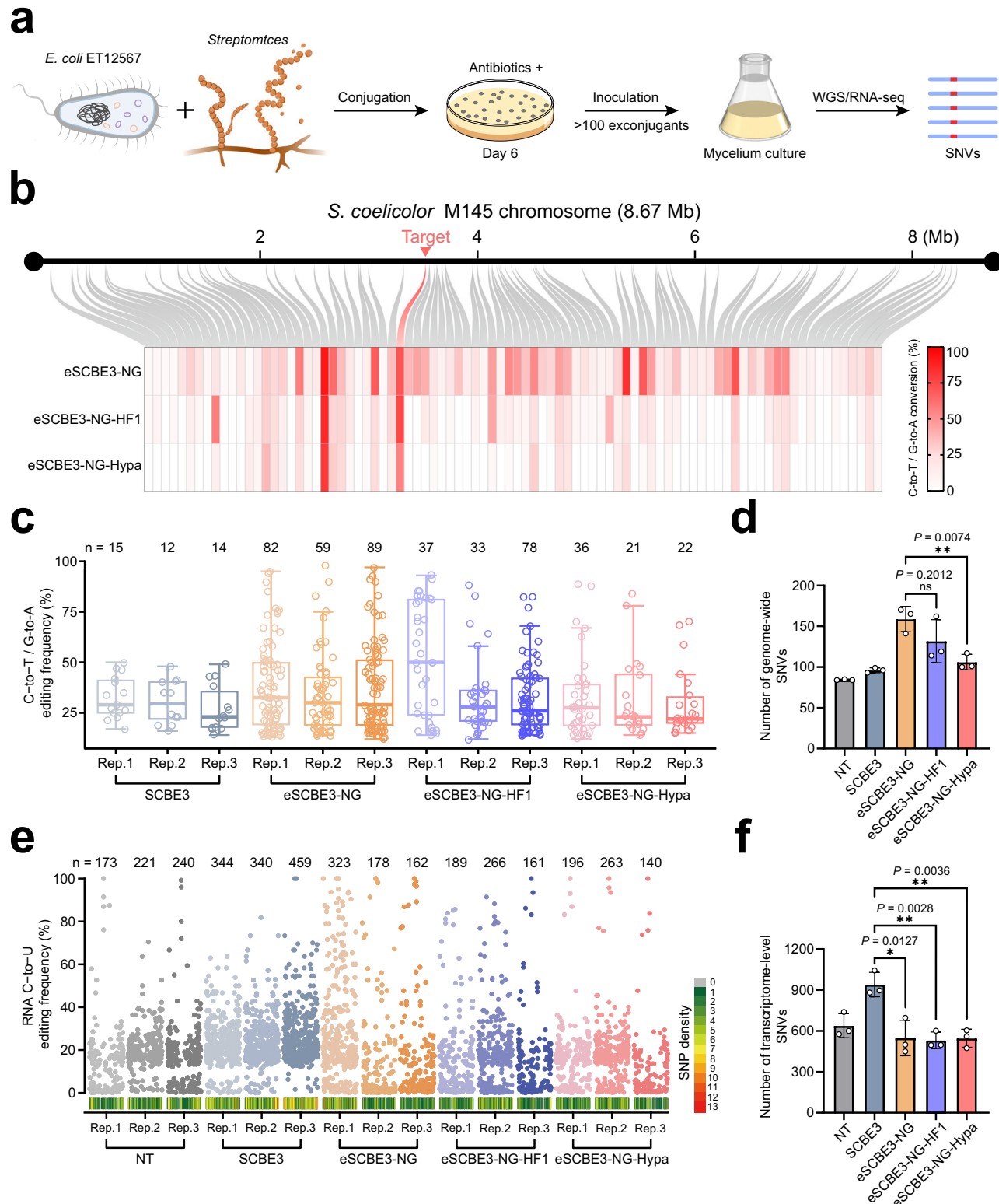

by eSCBE3-NG-Hypa were calculated to be up to 6.27 and 5.52, respectively, while SCBE3 displayed ratios of 1.04 and 1.91 (Fig. 5c). Moreover, at the PSP_EW9 site, the cognate-to-bystander editing ratio (C6: C5) was calculated to be up to 6.72. In comparison, SCBE3 displayed a ratio of 1 (Fig. 5c). These findings indicate that eSCBE3-NG-Hypa can more selectively edit a target C at positions 7 to 9 of the protospacer than SCBE3 and demonstrates less bystander activity than SCBE3.

Bioinformatics analysis of SCBE3 and eSCBE3-NGs in *S. coelicolor* M145 revealed numerous protospacers enabling the introduction of stop codons, containing at least one GC motif in the defined editing window from 4 to 8 (Fig. 3b). However, the widely used BE3 system typically exhibits inefficiency at these sites. Consistent with prior findings[13,19,35], it was observed that SCBE3 displayed a discrimination against the target C within the GC motif of PSP_EW1, PSP_EW2, and PSP_EW4 sites (Fig. 5d), particularly, the editing efficiency of the target

**Fig. 4 | Genome- and transcriptome-wide off-target effects induced by eSCBE3-NG-HF1 or eSCBE3-NG-Hypa. a** Scheme of the experimental workflow of WGS or RNA-seq in *S. coelicolor* M145. **b** Comparison of the detected DNA off-target C-to-T/G-to-A editing at the whole genome level. The off-target events within the population of over 100 exconjugants treated with eSCBE3-NG are distributed throughout the genome of *S. coelicolor* M145, indicated by the grey curve, while the on-target site is represented by the red curve. The heatmap illustrates the mean frequencies (*n* = 3 independent biological replicates) of off-target C-to-T/G-to-A conversions induced by eSCBE3-NG, alongside a comparison with eSCBE3-NG-HF1 and eSCBE3-NG-Hypa at the identified off-target sites of eSCBE3-NG. **c** The numbers and frequencies of total DNA off-target C-to-T/G-to-A editing induced by the indicated

BEs. Boxplot elements shown are the median (midline), the interquartile range of 25% and 75% percentiles (box boundaries). The whiskers mark the 5th and 95th percentiles. **d** Comparison of the total number of detected DNA SNVs induced by indicated BEs. Data are mean ± s.d. from three independent experiments. **e** Manhattan plot of RNA off-target C-to-U editing frequencies induced by indicated BEs. Each genome was separated into many bins (bin size is 0.1 Mb), the number of SNVs located within each bin was calculated and plotted. Different colors correspond to SNV density. **f** Comparison of the total number of detected RNA SNVs. Data are mean ± s.d. from three independent experiments. ns (not significant), *P* < 0.05, **P* < 0.01. The statistical test was two-sided and no adjustments were made for multiple comparisons.

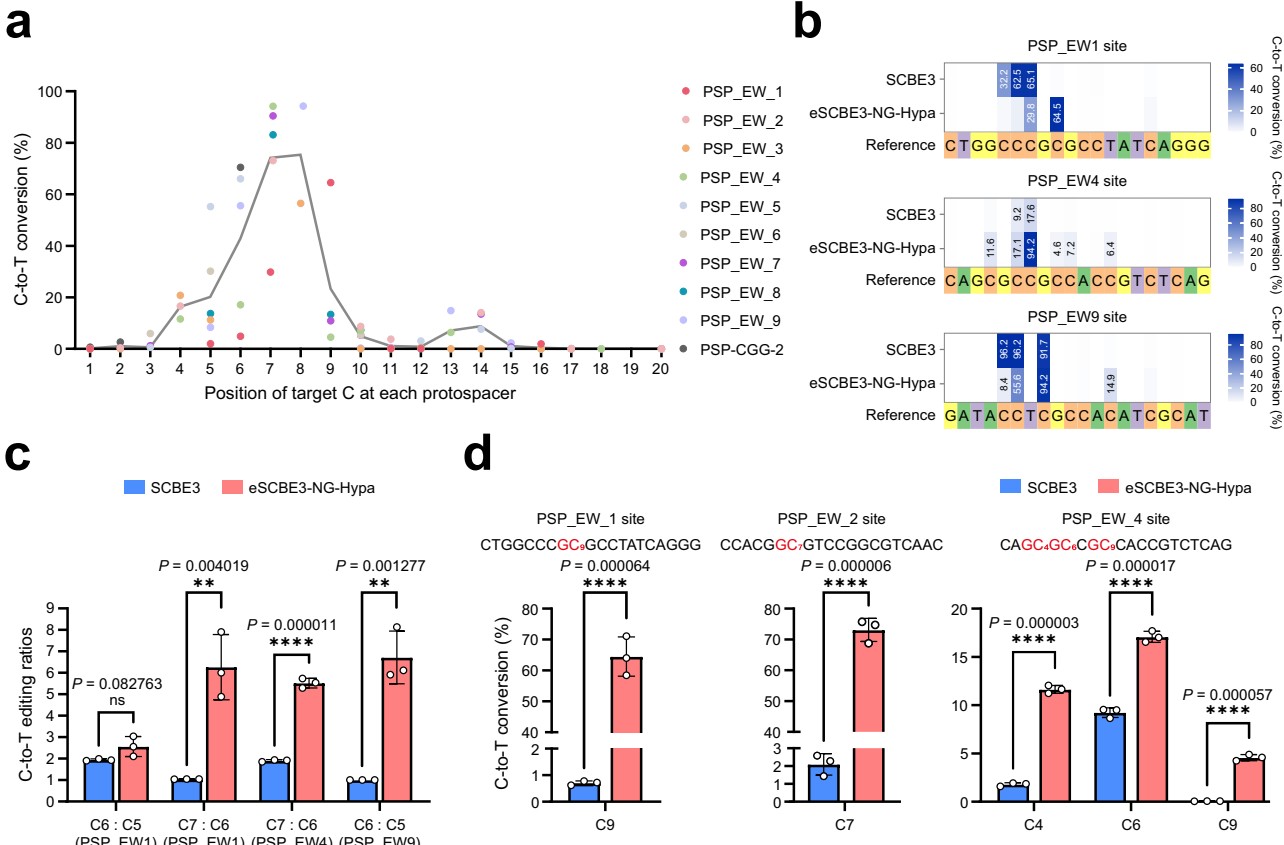

**Fig. 5 | Base editing specificity of eSCBE3-NG-Hypa across the protospacer regions. a** Scatterplots of average (*n* = 3) C-to-T editing of all cytosines in the protospacer compared to position in the protospacer for ten sites. Based on the average value of all scatter points at each protospacer position, broken line shows the base-editable ranges within the protospacer region by eSCBE3-NG-Hypa to define the editing window. **b** Heat maps are showing average C-to-T editing efficiencies of *n* = 3 independent biological replicates of SCBE3 and eSCBE3-NG-Hypa.

**c** C-to-T editing ratios of the cognate Cs within the protospacers shown in (**b**). Mean and s.d. shown for *n* = 3. **d** C-to-T editing efficiency and specificity of eSCBE3-NG-Hypa at GC motif. Bar figures show means and error bars representing s.d. of *n* = 3 independent biological replicates. ns (not significant), ***P* < 0.01, ****P* < 0.001, *****P* < 0.0001. The statistical test was two-sided and no adjustments were made for multiple comparisons.

C in GC9 of PSP_EW1 and GC7 of PSP_EW2 was minimal, measuring at 0.71% and 2.1%, respectively (Fig. 5d). In contrast, eSCBE3-NG-Hypa induced robust modifications with efficiencies reaching up to 64.51% and 73.09% at same target Cs, respectively, indicating that eSCBE3-NG-Hypa exhibits potent activity within GC motifs and is well-suited for gene editing in GC-rich *Streptomyces* genomes (Fig. 5d).

### Precise modification of PKS using eSCBE3-NG-Hypa

*Streptomyces* are prolific producers of antibiotics, primarily synthesized through PKS or non-ribosomal peptide synthetase (NRPS). Engineering the PKS/NRPS assembly lines represents a crucial approach for generating significant derivatives of bioactive natural products. Nevertheless, the intricate challenge arises from the

substantial homology within the highly repetitive domains of PKS/NRPS, rendering precise gene editing difficult. To assess the feasibility and precision of base editing within regions of high sequence homology, we try to apply eSCBE3-NG-Hypa to edit the repetitive domains of the PKS responsible for avermectins biosynthesis in the avermectins-high-producing industrial strains *S. avermitilis* 3-115[57].

In the biosynthetic assembly line of avermectins, six dehydratase (DH) domains are identified, in which DH7 must be inactive[58] according to the known colinearity of canonical modular PKS between multi-enzyme function and chemical structure (Fig. 6a and Supplementary Fig. 5). Alignment of the sequences of these six DH domains reveals a high homology, reaching up to 86.6% within their core regions (Fig. 6b). Notably, a conserved H×××G××××P motif, carrying part of the

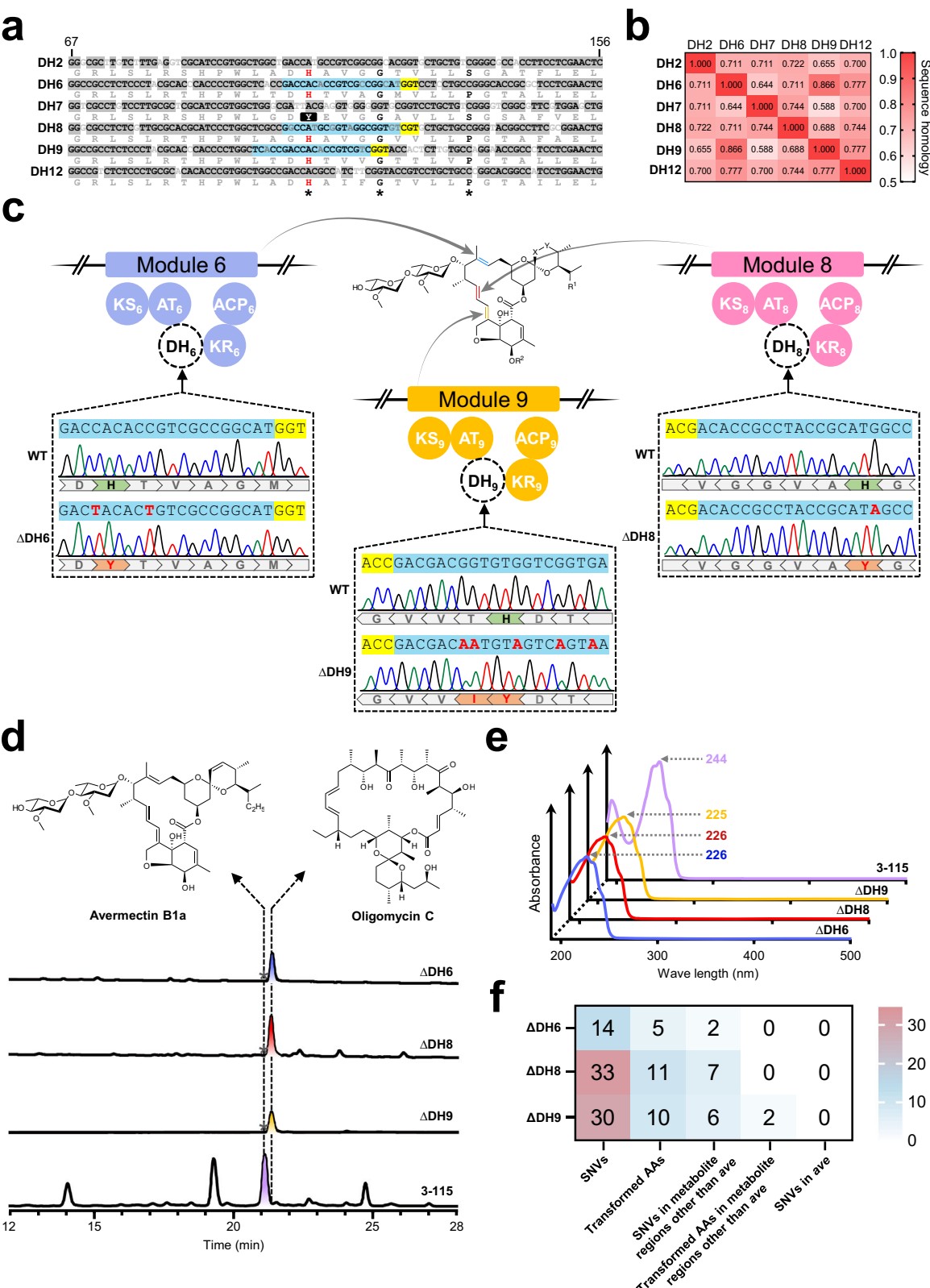

established His/Glu catalytic diad, is observed, where the His residue is an active site[59,60] (Fig. 6a). In contrast, the inactive DH7 domain features a distinct Y×××G×××S motif. Thus, we hypothesized that the substitution of histidine (His) with an uncharged tyrosine (Tyr) could lead to the deactivation of the DH7 domain. This alteration in amino acid corresponds to the codon change from CAT/CAC to TAT/TAC, a

modification achievable through cytosine base editing. Therefore, we proposed that the substitution of the active sites His in DH6, DH8, and DH9 domains with Tyr may also render these domains non-functional using eSCBE3-NG-Hypa, and corresponding mutants hold the potential to yield derivatives of avermectins. In light of eSCBE3-NG-Hypa's proficiency in recognizing NG PAM, three sgRNAs were designed to guide

**Fig. 6 | eSCBE3-NG-Hypa facilitates precise amino acid conversions within the modular repetitive PKS of *S. avermitilis*. a** Sequence alignments of the core region (67–156 bp) of DH domains within the avermectins PKS are presented. Conserved motif H×××G××××P is marked with black asterisks, wherein the active His is indicated in red, while Tyr at the same position within the inactive DH7 domain is highlighted in dark. Protospacers designed to induce the conversion of His to Tyr within domain DH6, DH8, and DH9 are marked with cyan shading, accompanied by the corresponding PAMs highlighted in yellow. Grey shading indicates the homologous region of DNA sequence. **b** The DNA sequence homology of the core region within DH domains shown in (**a**). **c** Base editing at protospacers shown in (**a**) was detected by Sanger sequencing. Chromatograms represent the editing events in *S. avermitilis* 3-115 and mutants ΔDH6, ΔDH8, and ΔDH9. The target His in domains DH6, DH8, and DH9 of avermectins PKS are indicated by green dovetail-shaped arrows, while the amino acid conversions observed in mutants ΔDH6, ΔDH8, and ΔDH9 are denoted by orange dovetail-shaped arrows. Protospacers and PAM sequences are highlighted in cyan and yellow, respectively. **d** HPLC analysis of fermentation extracts from control strain *S. avermitilis* 3-115 and mutants ΔDH6, ΔDH8, and ΔDH9 at λ = 245 nm. The dotted line indicates the presence of the same compounds, while the asterisks denote components that were not detected. Experiments were repeated three times independently with similar results. **e** UV/Vis spectra of compounds shown in **d**. **f** Genome-wide off-target evaluation of eSCBE3-NG-Hypa in mutants ΔDH6, ΔDH8, and ΔDH9. The heat map shows distribution of the nucleotide and amino acid changes against the reference genome of *S. avermitilis* 3-115.

the base editor to induce mutations that would replace the active site His codon (CAT/CAC) in the domains DH6, DH8, and DH9 with Tyr (TAT/TAC), respectively (Fig. 6c). Notably, it is not feasible to achieve the corresponding amino acid conversions in domains DH8 and DH9 using the BE3 system due to the absence of an indispensable NGG PAM. Sanger sequencing analysis revealed a target C/T conversion or overlapping peak in genomes of 33% exconjugants in group ΔDH6 (3 out of 9), 50% in group ΔDH8 (3 out of 6), and 43% in group ΔDH9 (3 out of 7) (Fig. 6a and Supplementary Fig. 6a–c). The observed C/T overlapping peaks in individual exconjugants can be attributed to the fact that the genomes of *Streptomyces* mycelium (up to 50) may share a common cytoplasmic compartment[4]. Candidate exconjugants have undergone plasmid curing to obtain a pure mutant. As anticipated, plasmid-cured mutants ΔDH6, ΔDH8, and ΔDH9 all demonstrated the complete substitution of the active site His with Tyr in the corresponding DH domains (Fig. 6c). HPLC and LC-ESI-HRMS analysis of the fermentation extracts showed the abolishment of avermectins production in all mutants (ΔDH6, ΔDH8, and ΔDH9) (Fig. 6d and Supplementary Fig. 7). Concurrently, a peak exhibiting distinct ultraviolet/visible (UV/Vis) absorption characteristics, with a maximum absorption wavelength of 225/226 nm, was detected, which differs from the 244 nm maximum absorption wavelength characteristic of avermectin B1a (Fig. 6d, e). However, subsequent $^1$H-NMR and LC-ESI-HRMS analyses confirmed the identity of the new peak as oligomycin C, another compound synthesized by the PKS of *olm* gene cluster within *S. avermitilis*[61,62] (Supplementary Table 1 and Supplementary Figs. 7, 8). Since the stringent substrate recognition process employed by PKS, we surmised that the downstream domains of the avermectins PKS, acting as gate-keepers[63], fail to recognize and accept uncanonical intermediate carbon chains due to incomplete catalysis by the inactive DH domains, resulting in the halt of carbon chain elongation, ultimately leading to the abolishment of avermectins production. Consequently, a metabolic shunt redirects resources from avermectins to oligomycins biosynthesis due to potential competition between the biosynthetic pathways[64]. This, in turn, leads to a substantial increase in the yield of oligomycins.

Furthermore, we naturally sought whether potential off-target effects induced by eSCBE3-NG-Hypa caused the inactivation of avermectins production. To examine it, we performed whole-genome re-sequencing on all mutants, including ΔDH6, ΔDH8, and ΔDH9. The results revealed that a limited number of 14, 33, and 30 potential SNVs along with 5, 11, and 10 amino acid alterations were detected in the genome of ΔDH6, ΔDH8, and ΔDH9, respectively (Fig. 6f and Supplementary Fig. 9a). However, no off-target site was detected in the *ave* gene cluster of the mutants ΔDH6, ΔDH8, or ΔDH9 genomes, suggesting that eSCBE3-NG-Hypa induced precise gene editing within the highly repetitive modular PKS genes (Fig. 6f and Supplementary Fig. 9b). The reason of the interruption of avermectins production should be attributed by the specific alteration of the key amino acid.

In summary, our results highlight the capability of eSCBE3-NG-Hypa to facilitate the accurate and efficient replacement of critical amino acids within highly repetitive modular PKS genes, showing its significant potential in modifying biosynthetic pathways to explore bioactive natural products.

## Applying eSCBE3-NG-Hypa to metabolic engineering in *S. avermitilis*

To expand the capabilities of eSCBE3-NG-Hypa in metabolic engineering, we sought to employ eSCBE3-NG-Hypa to further enhance the metabolic level of avermectins in an industrial high-yield strain *S. avermitilis* 3-115. Building upon our previous observations regarding the competitive relationship between the BGCs *ave* and *olm*, we designed a sgRNA-olm to introduce a premature stop codon in the *olmA1* region of the *olm* gene cluster (Fig. 7a). Additionally, we designed a sgRNA-pte to introduce a premature stop codon in the *pteA1* of *pte* gene cluster known for encoding PKS and producing detectable polyene macrolide filipins[61] (Fig. 7a). Furthermore, simultaneous disruption of both the *pte* and *olm* gene clusters was also devised, with the anticipation of achieving a cumulative increase in avermectins production.

Sanger sequencing analysis revealed a target C/T conversion or overlapping peak in genomes of 100% exconjugants in group Δolm (5 out of 5), 100% in group Δpte (1 out of 1), and 60% in group ΔolmΔpte (3 out of 5) (Fig. 7a and Supplementary Fig. 10a–c). After plasmid curing, we observed the anticipated amino acid mutation from Trp codon (TGG, corresponding to the editable CCA in the non-coding strand) to a stop codon (TAG/TAA) in mutants Δolm, Δpte, and ΔolmΔpte (Fig. 7a). As the fermentation results shown, the introduction of a premature stop codon in *olmA1* or *pteA1* led to the abolishment of oligomycins or filipins production (Supplementary Fig. 11). Prominently, among the four tested strains, Δolm and ΔolmΔpte demonstrated an accelerated rate of antibiotic production compared to the control strain (Fig. 7b). Meanwhile, the Δolm strain exhibited the highest 4.45-fold increase in avermectin B1a production, while the Δpte mutant showed a substantial 4.08-fold increase (Fig. 7b). Intriguingly, despite the dual inactivation of *olm* and *pte* in the ΔolmΔpte strain, it demonstrated only a modest 1.39-fold increase in avermectin B1a production (Fig. 7b).

This suggests a complex regulation in avermectins synthesis, where interwoven metabolic networks play a role in influencing the production of diversiform classes of secondary metabolites in *S. avermitilis*. The metabolic network flux and proteomics analysis of *S. avermitilis* suggest that the efficient production of avermectin B1a relies on the availability of correct precursors in the right ratios: 1 × methylbutanoyl-CoA + 5 × methylmalonyl-CoA + 7 × malonyl-CoA[65]. Given that the *ave*, *olm*, and *pte* gene clusters encode type 1 PKSs utilizing common building blocks or intermediates like malonyl-CoA, methylmalonyl-CoA, and propionyl-CoA, the disruption of either the *olm* or *pte* competitor pathway leads to the accumulation of avermectins precursors, thereby enhancing avermectins production. However, the excess accumulation of malonyl-CoA, methylmalonyl-CoA and propionyl-CoA may be inhibitory and toxic to the cell[66]. In line

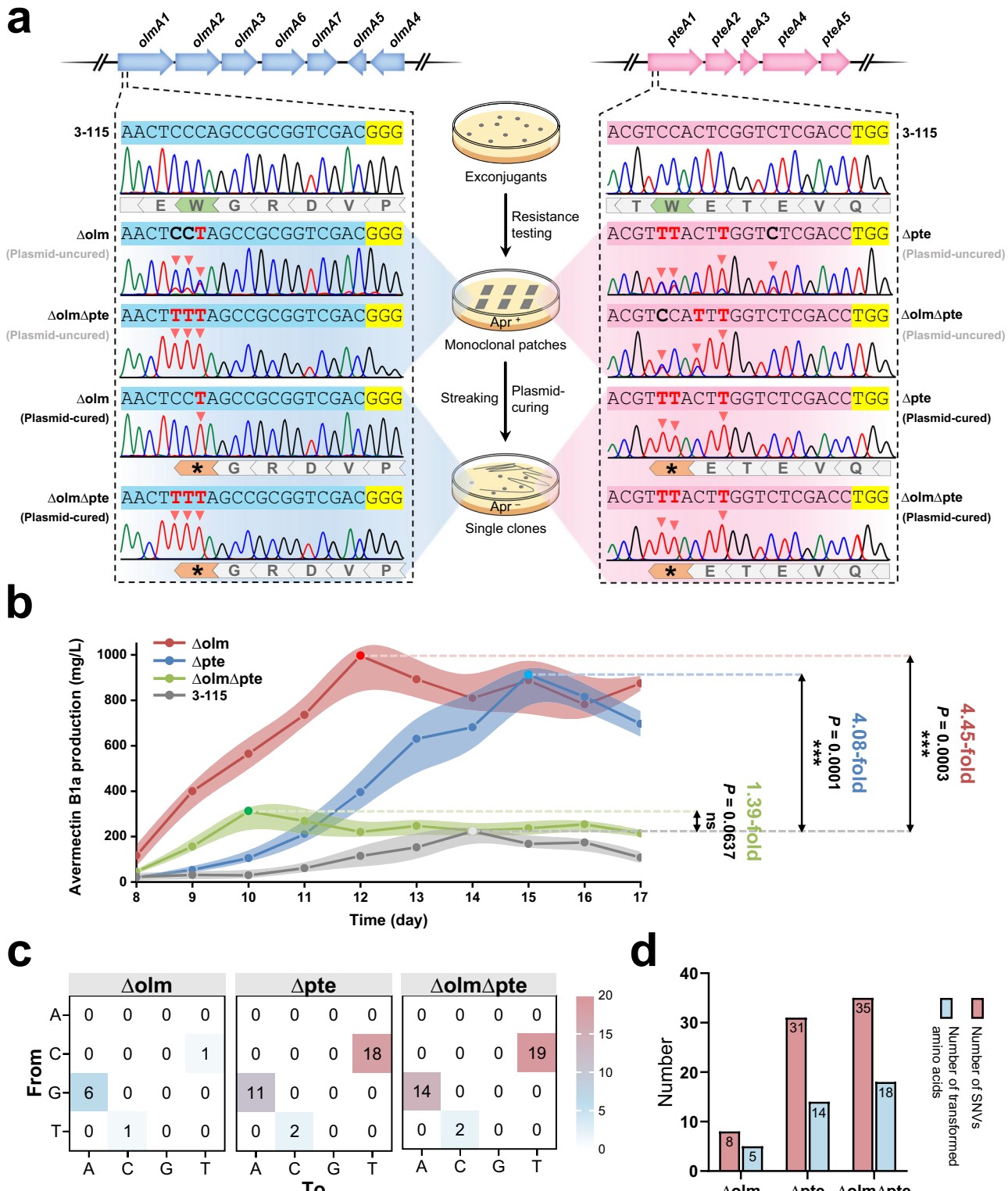

**Fig. 7 | Application of eSCBE3-NG-Hypa in metabolic engineering. a** The deactivation of *olm* and *pte* in *S. avermitilis* 3-115 via the introduction of premature stop codon by using eSCBE3-NG-Hypa. Blue arrows indicate PKS gene in *olm*, and red arrows indicate PKS gene in *pte*. Doted areas are the protospacers designed to generate premature stop codon in gene *olmA1* and *pteA1*, respectively. PAM is highlighted in yellow. The Sanger sequencing chromatograms illustrate the editing events observed in plasmid-uncured exconjugants and plasmid-cured mutants. The detected base conversions are marked with red triangles. The target tryptophan (W) residue is highlighted by green dovetail-shaped arrows, whereas the intended stop codons are denoted by black asterisks highlighted in orange dovetail-shaped arrows. **b** Line chart with error band shows avermectin B1a production of mutants ΔDH6, ΔDH8, and ΔDH9. Mean and s.d. shown for *n* = 3 biologically independent samples. The statistical test was two-sided and no adjustments were made for multiple comparisons. **c** Genome-wide off-target evaluation of eSCBE3-NG-Hypa in mutants Δolm, Δpte, and ΔolmΔpte. Heat map shows distribution of the nucleotide changes against the reference genome of *S. avermitilis* 3-115. **d** The bar chart displays the numbers of SNVs shown in (**c**) and transformed amino acids in mutants Δolm, Δpte, and ΔolmΔpte.

with the prior study[66], we demonstrated a delay in the growth of *S. avermitilis* 3-115 and its corresponding mutants when cultured in TSB medium supplemented with propionate compared to the group without propionate (Supplementary Fig. 12). We hypothesized that metabolites derived from propionate, specifically propionyl-CoA and methylmalonyl-CoA, induce growth inhibition in *S. avermitilis* 3-115 and related strains, constraining the production of secondary metabolites of *Streptomyces*.

Additionally, we investigated the off-target effects induced by eSCBE3-NG-Hypa in genomes of the mutant strains Δolm, Δpte, and ΔolmΔpte. WGS analysis revealed a limited number of 8, 31, and 35 SNVs in the genome of Δolm, Δpte, and ΔolmΔpte, respectively, with the primary mutation type being C-to-T/G-to-A (Fig. 7c, d and Supplementary Data 4). Meanwhile, less amino acid alteration caused by detected SNVs was observed with the number of 5, 14, and 18 in Δolm, Δpte, and ΔolmΔpte, respectively (Fig. 7d). Of note, no additional unexpected stop codon was introduced in the metabolic regions of the genomes of Δolm, Δpte, and ΔolmΔpte (Supplementary Fig. 9c), suggesting that eSCBE3-NG-Hypa exhibited an acceptable off-targeting profile, and the inactivation of the target competitor gene cluster primarily contributed to the enhancement of avermectins production in the mutant strains Δolm, Δpte, and ΔolmΔpte. These results signify that the inactivation of gene clusters within competitor pathways, facilitated by eSCBE3-NG-Hypa, represents a potent approach to attain precise gene control and enhance product synthesis.

## Discussion

The applications of genome editing in *Streptomyces* necessitate the development of base editors with a broad targeting scope and minimal off-target effects. Here, we engineered and characterized 11 cytosine base editors with relaxed PAM tolerances, high-fidelity properties, or a combination of both, of which eSCBE3-NGs displayed considerable editing efficiency at target sites featuring NG PAM. Although the recently reported SpCas9 variant SpRY allows for targeting of sites with preferred NRN (where R is A or G) and NYN (where Y is C or T) PAMs, it leads to the recognition of more off-target sites[39]. Furthermore, the results of bioinformatics analysis indicate that up to 98.72% of genes in the genome of *S. coelicolor* M145 could be introduced a premature codon, suggesting that a substantial number of NGN PAMs distributed across the *Streptomyces* genome render SpCas9-NG a fitting choice for diverse applications with comparatively low off-target risks in *Streptomyces*.

We demonstrated that a high GC content within the protospacer sequence can severely erode the editing efficiency for the widely used BE3 system. A plausible explanation is that SpCas9 encounters difficulty in DNA strand separation and R-loop formation at protospacers with high GC content. However, the eSCBE3-NGs display remarkable activity at these sites, which can be attributed to the high activity of hAPOBEC3A(Y130F). This suggests that eSCBE3-NGs are well-suited for applications in high GC content genomes, such as those found in *Streptomyces*.

To comprehensively evaluate the safety profile of eSCBE3-NGs, we conducted whole-genome re-sequencing and RNA-seq in a large population of exconjugants. The results demonstrated that eSCBE3-NG-Hypa induces low levels of both DNA and RNA off-target SNVs. Although SCBE3 induced the fewest C-to-T/G-to-A conversions among all experimental groups (Fig. 4c), its lower on-target editing efficiency and limited targeting scope should be taken into account. SCBE3 exhibited a mean C-to-T editing rates of 45.7% at the on-target site PSP_CGG-4, whereas that of eSCBE3-NG-Hypa was 56.3% (Supplementary Data 3). Additionally, we demonstrated that SCBE3 discriminates against target sites characterized by non-NGG PAM, high GC content or GC motif, resulting in significantly fewer editable sites across the whole genome compared to eSCBE3-NG-Hypa.

Considering these factors, along with the concern regarding RNA off-target effects of SCBE3 (Fig. 4e, f), we recommend using eSCBE3-NG-Hypa in most application scenarios to achieve high-efficiency and -fidelity in base editing.

We observed a few potential off-target sites with high C-to-T/G-to-A editing frequency (>50%) induced by eSCBE3-NG-Hypa at the whole-genome level. Further analysis revealed that these high-risk off-target sites mostly contained base pair mismatches in the PAM-distal position of protospacers (Supplementary Fig. 13a). This can be accredited to the high tolerance of SpCas9 orthologues towards off-target sites with mismatches located distal from the PAM site[67,68], which has posed challenges in minimizing off-target effects. Nevertheless, a recently reported SpCas9 variant, SuperFi-Cas9, exhibited the ability to discriminate off-target DNA substrates containing mismatches at PAM-distal positions without sacrificing DNA cleavage activity[69]. In pursuit of a higher-fidelity base editor, we constructed eSCBE3-NG-SuperFi, eSCBE3-NG-HF1-SuperFi, and eSCBE3-NG-Hypa-SuperFi, all derived from the eSCBE3-NGs (Supplementary Fig. 13b). Unfortunately, all base editors incorporated amino acid substitutions of SuperFi-Cas9 displayed low editing efficiency, ranging from 0% to 7% editing (Supplementary Fig. 13c), suggesting that SuperFi-Cas9 may not be compatible with cytosine base editors, in agreement with previous findings[70]. As SuperFi-Cas9 has shown unaffected on-target activity exclusively in vitro[69], it is comprehensible that SuperFi-Cas9 has low activity in *Streptomyces*. Even so, in challenging circumstances necessitating higher fidelity, we recommend the use of eSCBE3-NG-HF1-Hypa at the price of on-target activity or eSCBE3-NG-Hypa in combination with engineered sgRNAs, such as truncated sgRNA[71] or hp-sgRNA[72].

To comprehensively assess the off-target effects induced by BEs, WGS serves as the gold standard. Tong et al. conducted WGS on *S. coelicolor* treated with CBE3 and identified 22 and 12 SNVs in two pure mutants, respectively[35]. Although eSCBE-NG-Hypa in this study induced an average of 21.7 SNVs across three independent biological samples from *S. coelicolor*, it is important to note that our genomic samples were derived from a mixture of over 100 exconjugants, with SNV frequencies ranging from 12% to 89% (Supplementary Data 3). Our approach has contributed to the detection of more potential SNVs, showing a comprehensive off-target profile induced by BEs. When assessing the applicability of a base editing tool, particularly for highly sequence-similar coding regions such as PKS/NRPS, ensuring its safety is crucial. As our unpublished preprint study, the widely used CBE3 system induced unintended off-target effects at the *aveA4* gene, despite the sgRNA being programmed to target the *olmA1* gene in *S. avermitilis*[73]. This ultimately led to the cessation of avermectins production. However, the eSCBE3-NG-Hypa developed in this study enabled precise base editing using the same sgRNA without any unexpected SNVs in the *ave* or *olm* gene clusters, as confirmed by whole-genome sequencing data for mutants Δolm. (Supplementary Fig. 9c). In addition, as indicated by previous research, the three-dimensional organization of the chromosome in the host strain is correlated with alterations in the expression of biosynthetic clusters[74]. Inactivation of gene cluster by introducing a premature stop codon via CBE, as opposed to the conventional insertion or deletion of large fragments, represents a more conservative and recommended approach for gene cluster knock-out without inducing changes in the three-dimensional structure of the genome.

Taken together, the engineered cytosine base editor, eSCBE3-NG-Hypa, presented here exhibits a broad targeting scope, high-fidelity, low bystander activity, and does not discriminate against GC motif or sites with high GC content, thus could widen the potential applications of precision genome editing in *Streptomyces*. We anticipate that eSCBE3-NG-Hypa will facilitate advancements in the modification of biosynthetic pathways and metabolic engineering.

## Methods

### Bacterial strains, culture conditions and reagents

*E. coli* DH10B was used for plasmid construction. *E. coli* ET12567 harboring pUZ8002 was used as the donor for intergeneric conjugation between *E. coli* and *Streptomyces*. *E. coli* strains were maintained in 2 × TY medium[4] at 37 °C, antibiotics (apramycin 25 μg/mL, ampicillin 100 μg/mL, chloramphenicol 25 μg/mL, and kanamycin 25 μg/mL) were added into the medium when required. *S. coelicolor* was cultivated on SFM agar medium[4] at 28 °C for sporulation, while *S. avermitilis* was cultivated on ISP4 agar medium[75] at 28 °C for sporulation. All chemical reagents and antibiotics were purchased from Sigma Aldrich and Sangon Biotech. All molecular cloning reagents were purchased from New England BioLabs and QIAGEN. All oligonucleotide synthesis and Sanger sequencing services were provided by Wuhan Tsingke. Bacterial strains are summarized in Supplementary Table 3.

### Plasmid construction

Plasmids and DNA oligos used in this study are listed in Supplementary Data 1. Base editor expression plasmids pSCBE3[73] was constructed using pYH7[50], a self-replicating *E. coli-Streptomyces* shuttle vector, as the parent plasmid. The DNA fragment containing point mutations of SpCas9-NG (R1335V/L1111R/D1135V/G1218R/E1219F/A1322R/T1337R) was synthesized by GenScript and amplified using the primer pair Fwd-SpCas9-NG and Rev-SpCas9-NG through polymerase chain reaction (PCR). Subsequently, this fragment was cloned with a linear fragment, obtained from the amplification of pSCBE3 using the primer pair Fwd-Vec-SCBE3 and Rev-Vec-SCBE3, through Gibson assembly to yield pSCBE3-NG. The substitutions characteristic of SpCas9-HF1 (N497A/R661A/Q695A/Q926A) and HypaCas9 (N692A/M694A/Q695A/H698A) were introduced by PCR at the portion of SpCas9 (D10A) in pSCBE3 or pSCBE3-NG, resulting in pSCBE3-HF1, pSCBE3-Hypa, pSCBE3-HF1-Hypa, pSCBE3-NG-HF1, pSCBE3-NG-Hypa and pSCBE3-NG-HF1-Hypa. For the construction of peSCBE3, a DNA fragment encoding hAPO-BEC3A(Y130F) was synthesized by GenScript and amplified using the primer pair Fwd-A3A(Y130F) and Rev-A3A(Y130F) through PCR. Subsequently, this fragment was cloned with linear fragments obtained from the amplification of pSCBE3-NG, pSCBE3-NG-HF1, pSCBE3-NG-Hypa, and pSCBE3-NG-HF1-Hypa using the primer pair Fwd-Vec-eSCBE3 and Rev-Vec-eSCBE3, employing Gibson assembly to obtain peSCBE3-NG, peSCBE3-NG-HF1, peSCBE3-NG-Hypa and peSCBE3-NG-HF1-Hypa. For the construction of SuperFi-Cas9 based base editors, a DNA fragment synthesized by GenScript and amplified using primer pair Fwd-SuperFi-Cas9 and Rev-SuperFi-Cas9 was cloned with linear fragments obtained from the amplification of peSCBE3-NG, peSCBE3-NG-HF1, peSCBE3-NG-Hypa using the primer pair Fwd-Vec-SuperFi-Cas9 and Rev-Vec-SuperFi-Cas9, employing Gibson assembly to obtain peSCBE3-NG-SuperFi, peSCBE3-NG-HF1-SuperFi, and peSCBE3-NG-Hypa-SuperFi.

### sgRNA design and cloning

The sgRNAs for CBE were designed using the Benchling web tool (https://benchling.com/), in which the criteria for scoring base editing guide sequence were incorporated according to the in vitro activity of BE1. Because most sgRNAs used in this study needs to be cloned individually and connected in series, the MSCC was used for all sgRNA cloning to share primers. Cloning steps are described as followings: (1) Linearize the aimed BE plasmid with *Xba*I and *Nhe*I. (2) PCR amplifies the inserts using any of the MSCC containing BE plasmid as template, and introduce the 20 bp guide sequence, and the 20 bp vector overlaps by the 5′ end of the primers. (3) Ligate the inserts and the *Xba*I-*Nhe*I linearized BE plasmid by Gibson Assembly, then check the recombinants by colony PCR and Sanger sequencing.

### Interspecies conjugation

Conjugation experiment was undertaken following the standard protocol described previously[4]. Briefly, the plasmid donors *E. coli* ET12567/pUZ8002/BE plasmid were grown to an OD$_{600nm}$ of 0.4-0.6. Cells were pelleted by centrifugation at 2400 g for 5 min, washed twice in 2 × TY broth, and resuspended in 100 μL of 2 × TY. *Streptomyces* spores were washed twice in TES buffer (0.05 M, pH 8.0), followed by resuspension in 5 mL TES buffer and subsequent incubation at 50 °C for 10 min to induce germination. An equal volume of 2 × TY broth and 10 μL 5 M CaCl$_2$ were added and the mixture was incubated at 37 °C for 2–3 h with shaking at 220 rpm. The germinated spores were pelleted by centrifugation as stated above, resuspended in 100 μL of 2 × TY broth. *E. coli* cells and the prepared spores were mixed and spread onto SFM agar plate containing 10 mM MgCl$_2$ for *S. coelicolor*, while ISP4 agar plate containing 30 mM MgCl$_2$ for *S. avermitilis*. The conjugation plates were incubated for 12 h at 28 °C, then the surfaces of the plates were overlaid with 1 mL of sterile water containing nalidixic acid (final concentration is 25 μg/mL) and the apramycin (final concentration is 25 μg/mL) for both *S. coelicolor* and *S. avermitilis*.

### Base editing analysis by next-generation sequencing (NGS)

To evaluate the editing efficiency in a large base population, the genomic DNA was directly extracted from the mixture of over 100 exconjugants scraped out from the conjugation plates after overlaying with apramycin for 7 days. Target genomic sites were PCR amplified by high-fidelity DNA polymerase Phanta Max Super-Fidelity DNA Polymerase (Vazyme) with primers flanking 8-bp barcodes at the 5′ end. The PCR primers used for NGS are listed in Supplementary Data 1. PCR products were verified through agarose gel electrophoresis and purified using Universal DNA Purification Kit (Tiangen Biotech). Amplicons with various barcodes were mixed and sequenced on an Illumina MiSeq instrument according to the manufacturer's protocols (GENEWIZ). The quality of raw sequencing reads was evaluated using fastp software[76], and those with a quality score below 15 were discarded. Adapters were trimmed via PANDAseq[77], and Fastq multx[76] demultiplexed paired sequences. CRISPResso2[78] was used to calculate the editing efficiency of CBE. Base substitution frequencies were calculated by dividing base substitution reads by total reads, and corresponding results of base substitution frequencies are listed in Supplementary Data 5.

### Off-target effects evaluation through whole genome re-sequencing

After the conjugation plates were overlaying with apramycin for 6 days, over 100 exconjugants were scraped out and inoculated into 30 mL fresh TSBY broth (3% tryptone soy broth, 0.5% yeast extract, 10.3% sucrose) without antibiotics and incubated at 28 °C and 220 rpm on a rotary incubator for 2–3 days. Mycelial fragments were harvested by centrifugation at 6200 *g* for 10 min, washed twice in sterile water, and pelleted by centrifugation. Collected mycelia were used for genomic DNA extraction. Next-generation sequencing library preparations were constructed following the manufacturer's protocol (GENEWIZ). The qualified libraries were sequenced pair-end PE150 on the Illumina HiseqXten/Novaseq/MGI2000 System. Using fastp[76] removed the sequences of adaptors, PCR primers, N base more than 14, and Q20 lower than 40%. Pipeline of Sentieon was used to map clean data to *S. coelicolor* genome (NC_003888.3) or *S. avermitilis* 3-115 genome deposited in the NCBI Sequence Read Archive database under accession number PRJNA1066248, remove duplication and call SNVs[79]. Found SNVs with sequencing depth of >10 were counted.

### RNA-seq experiments

Procedures of sample preparation are the same as that in whole genome re-sequencing. Total RNA was extracted from the mycelia

according to the manufacturer's instructions of TRIzol (Invitrogen), and genomic DNA was removed using DNase I (TaKaRa). RNA degradation and contamination were monitored on 1% agarose gels. Then RNA quality was determined by 2100 Bioanalyser (Agilent Technologies) and quantified using the ND-2000 (NanoDrop Technologies). Only high-quality RNA sample was used to construct the sequencing library. RNA purification, reverse transcription, library construction, and sequencing were performed at Majorbio Bio-pharm Biotechnology Co., Ltd. The RNA library was sequenced using Illumina NovaSeq 6000 sequencer in 2 × 150 bp read length. The raw paired-end reads were trimmed and quality was controlled by fastP with default parameters. Then clean reads were separately aligned to the *S. coelicolor* genome (NC_003888.3) with orientation mode using HISAT2[80] software. StringTie assembled the mapped reads of each sample in a reference-based approach[81]. SNVs calling analysis was performed by Sentieon[79]. Found SNVs with sequencing depth of >10 were counted.

### Fermentation conditions

Avermectins-high-producing industrial strains *S. avermitilis* 3-115 and corresponding mutants were cultivated in ISP4 agar medium for rejuvenation. Mycelium from ISP4 agar medium were transferred into 250 mL flasks containing 30 mL TSB seed medium (30 g tryptone soya broth per liter) and incubated at 28 °C and 220 rpm on a rotary incubator for 2 days. 500 mL fermentation mediums were then inoculated with 5% seed cultures and incubated on a rotary shaker at 220 rpm at 28 °C for 10 days. The fermentation medium contained (per liter) 140 g of corn starch, 0.1 g of α-amylase, 28 g of soy flour, 10 g of yeast extract, 0.022 g of $Na_2MoO_4 \cdot 2H_2O$, 0.0023 g of $MnSO_4 \cdot H_2O$, 0.25 g of $(NH_4)_2SO_4$, 0.02 g of $CoCl_2$, and 0.8 g of $CaCO_3$. The medium pH was adjusted to 7.5 with NaOH before autoclaving.

### Isolation and detection of secondary metabolites of *S. avermitilis*

For avermectins extraction, the 5 mL fermentation broth from three biological replicates was diluted by mixing with 25 mL methanol and sonicated for 30 min. After centrifuging (13,800 *g*, 10 min), the supernatants were evaporated under reduced pressure after centrifugation. Subsequently, the concentrated products were dissolved in 1 mL methanol again and subjected to HPLC (SHIMADZU) and LC-ESI-HRMS (Thermo) analysis after filtration through a 0.22 µm Nylon66 membrane. Avermectins were detected by HPLC using a Phenomenex Luna C18 column (5 µm, 250 × 4.6 mm) at a flow rate of 1.0 mL/min using a mobile phase of (A) 0.1% formic acid in water and (B) acetonitrile. The separation gradient is as follows: 0–5 min, 65% B; 5–25 min, 65–100% B; 25–26 min, 100–65% B; 26–30 min, 65% B. Detection by LC-ESI-HRMS was on a Thermo Electron LC-ESI-HRMS using positive mode electrospray ionization fitted with a Phenomenex Luna C18 column (5 µm, 250 × 4.6 mm) at a flow rate of 0.8 mL/min using the same mobile phase and separation gradient with HPLC detection. The mass spectrometer was set to full scan (from 500 to 1250 m/z). The data were collected and processed using Thermo Xcalibur software 3.0.63 for peak alignment and formula weight layout.

### NMR analysis of detected compounds

1D ($^1$H) NMR spectra were collected on an Agilent-NMR-VNMRS 600 spectrometer (Supplementary Table 1 and Supplementary Fig. 8). Chemical shifts were reported in ppm using tetramethylsilane as an internal reference, and coupling constants were reported in Hz. NMR data processing was performed using MestreNova software. The molecular formula of compound was determined according to their LC-ESI-HRMS data.

### Bioinformatics analysis of base editing in *S. coelicolor* M145

A python script was coded to identify protospacers capable of introducing premature codons in the *S. coelicolor* M145 genome

(Supplementary Note 1)[82]. Protospacers with NGN PAMs were selected for SpCas9-NG-based base editing, while those with NGG PAM were targeted for SpCas9-based editing. The editing window was defined as 4–8 nucleotides at the PAM-distal position of the protospacer.

### Statistics and reproducibility

Statistical analysis was performed using GraphPad Prism 10.1.2 software using a two-tailed *t*-test analysis of variance hypothesis. Significant differences are marked as $*P < 0.05$, $**P < 0.01$, $***P < 0.001$, $****P < 0.0001$. All data are presented as mean ± s.d. or mean ± s.e.m. The number of biologically independent samples for each panel was three unless otherwise stated in the figure legends.

### Reporting summary

Further information on research design is available in the Nature Portfolio Reporting Summary linked to this article.

## Data availability

The NGS, WGS, and RNA-seq data for *S. coelicolor* M145 generated in this study have been deposited in the NCBI Sequence Read Archive database under accession code PRJNA1064385. The WGS data for *S. avermitilis* 3-115 and its derivative mutants generated in this study are available in the NCBI Sequence Read Archive database under accession code PRJNA1066248. Plasmids used in this study have been deposited to Addgene for distribution. The corresponding Addgene IDs are listed in Supplementary Data 1. All other data supporting the findings of this study are included in the published article and Supplementary Information. Requests for any additional information can be made to the corresponding authors. Source data are provided in this paper.

## Code availability

A Python script for identifying the protospacers enabling the introduction of premature codons in *S. coelicolor* M145 genome can be found in Supplementary Note 1 or at https://doi.org/10.5281/zenodo.11579018[82].

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

## Acknowledgements

We acknowledge the financial support provided by the National Key R&D Program of China (to Y.S., grant no. 2018YFA0903200), International Cooperation and Exchange of the National Natural Science Foundation of China (to Y.S., grant no. 31920103001), which made this research possible.

## Author contributions

Y.S. and J.W. conceived the study, designed the experiments, and wrote the manuscript; J.W., K.W., Z.Z., and Q.M. conducted the experiments; Zh.D. conducted bioinformatics analysis; G.S. analyzed the NMR data; J.W., K.W., Zh.D., Z.Z., G.S., Q.M., F.Z., Zi.D. and Y.S. contributed to data analysis and discussion.

## Competing interests

The authors declare no competing interests.
