## [Peer Review File · Nature Communications]

Engineered cytosine base editor enabling broad-scope and high-fidelity gene editing in *Streptomyces*REVIEWER COMMENTS

Reviewer #1 (Remarks to the Author):

Wang et al. developed an engineered cytosine base editor (CBE) eSCBE3-NG-Hypa in the important industrial streptomyces via extensive characterization and screening work. eSCBE3-NG-Hypa was established based on SCBE3 (codon-optimized of CBE3 for Streptomyces), in which the nCas9 (D10A) combines multiple excellent traits of Cas9 featured by high-fidelity and an expansive recognition scope, and the deaminase rAPOBEC1 was replaced with hAPOBEC3A(Y130F) with high activity in regions marked by GC motifs. Compared with the current base editors in Streptomyces, the engineered CBE eSCBE3-NG-Hypa shows a broad editing scope, high-fidelity, and low bystander activity, and does not discriminate against GC motif or sites with high GC content. Particularly, this editor could achieve base editing of DNA with up to 85% GC content. I believe it could widen the potential applications of precision genome editing in Streptomyces. This research work belongs to integrated innovation, with rigorous work design, reliable and detailed data, thorough analysis, and high quality charts. Overall, this is a high-quality research paper suitable for publication in this high-ranked journal. I have some comments as follows.

1. Abstract, Line 6: this sentence should be rephrased; in addition, "active histidine", "inactive tyrosine", in which "active" and "inactive" should be removed. Similarly at Page 28, "active His to inactive Tyr". Amino acids cannot be classified as active or inactive.
2. Page 24, second paragraph, "100% in group Δ pte (1 out of 1)", why did the author just analyze one exconjugant? or only 1 exconjugant was obtained from conjugation transfer experiment? Please briefly explain.
3. Fig7a legend. "The sanger sequencing chromatograms illustrate the editing events observed in exconjugants containing uncured plasmids and plasmid-cured mutants", please clearly indicate in the figure which chromatograms are the editing events from exconjugants containing uncured plasmids, and which chromatograms are from plasmid-cured mutants.
4. It's interesting to find that either of inactivation of olm and pte resulted in over 4-fold increase in avermectin B1a production; however, inactivation of both olm and pte only led to 1.39-fold increase in avermectin B1a titer. The authors performed WGS analysis of the mutants and they found no unexpected stop codon was introduced in the metabolic regions of the genome of Δ olm Δ pte mutant. I wonder whether there are amino acid alterations by detected SNVs in the avermectin biosynthetic genes or known regulatory genes involved in avermectin biosynthesis. Please check it.

Reviewer #2 (Remarks to the Author):

The work describes an advancement on the use of cytosine base editors in Streptomyces, which has previously proved challenging due to their high GC content. It is undoubtedly an advance on current methods, improving the options for PAM recognition, increasing efficiency and reducing off target activity. They then use their new methodology to introduce point mutations into avermectin producing streptomyces strains, first in an attempt to make analogues and then to increase production. This latter part is not particularly ground breaking, as it is well known that blocking production of one natural product can switch metabolic flux towards another, but is useful. The data is comprehensive, conclusions valid and methods sufficient.

What would have be useful to include is a more thorough analysis of how well the exemplar experiments worked with previous best methods. A minor comment in the discussion 'As shown in our prior study, the widely employed BE3 induced unintended off-target effects, ultimately leading to the cessation of avermectin production' suggests this has been done but this references an article that has not undergone peer review and it's hard to make a direct comparison. Similarly in the section 'DNA and RNA off-target evaluation of eSCBE3-NG-HF1 and eSCBE3-NG-Hypa' the control using scBE3 was only discussed for the transcriptomics assessment, not the initial experiment monitoring on-target

efficiency. Without these analyses its hard to judge whether the improved methodologies truly increase the applicability to an extent sufficient to warrant publication in nature communications.

My only other minor complaint is that there are an awful lot of acronyms in the article which makes it quite hard to follow which plasmid is which, especially for an article aimed at a general audience. Figure 1 attempts to do this pictorially but it would be useful to show functionally what all the various mutations actually do. At the moment the reader has to look at the figure, check what the colours are, find the corresponding mutations in the figure legend, then check the manuscript to see what these mean. A more succinct summary would be useful – or better yet, a less complicated naming system.

Reviewer #3 (Remarks to the Author):

This manuscript focuses on editing tool development of *Streptomyces*, which is known a one of the most prolific producers of antibiotics. By introducing the advance of cytosine base editors, the authors developed a more efficient base editors, namely eSCBE3-NG-Hypa, targeting the *Streptomyces* genome. The authors addressed the problems encountered by the present base editors in their application to *Streptomyces*, specifically the high GC content of its genome that affects the editing range and accuracy of CBE3.

The noteworthy results are that this study combined different cytidine deaminase and Cas9 mutants to finally generate an excellent cytosine base editor eSCBE3-NG-Hypa . Using this tool, the authors successfully introduce stop codons into multiple secondary metabolite synthesis gene clusters, achieving high production of avermectin in *S. avermitilis* . The article follows a clear logical structure encompassing problem finding and problem solving and its engineering application. Moreover, it effectively addresses limitations in CBE for *Streptomyces* genome editing while improving efficiency and fidelity. Therefore, this work is exciting in streptomyces community. However, considering all the cytidine deaminase and Cas9 mutants used in this work were previously used ones, I suggested the mechanism behind the excellent performance of eSCBE3-NG-Hypa should be further explored in this work.

Comments:

1. In Section 5 of the Results, while Fig. 4c did not provide a comparison of off-target rates for SCBE3 at the DNA level, the evaluation of SCBE3 was conducted at the RNA level in Fig. 4e. It would be valuable to investigate whether the tool developed in this study also exhibits an increase in off-target rates when utilizing Cas9 mutants to expand targeting range. If indeed the off-target rate decreased as depicted in Fig. 4e, it is important to understand the underlying reason.

2. Although t the application of eSCBE3-NG-Hypa in metabolic engineering of avermectin production demonstrated its powerful, a systematic analysis of the metabolic networks and resource allocation of different genetic modifications would provide a clearer explanation of the reasons for the distinct increases in Fig. 7b. Given the unexpected impact of $\Delta\text{olm}\Delta\text{pte}$ double mutant strain on avermectin titer and production rate in Figure 7b, I suspect there may be some unknown influence by eSCBE3-NG-Hypa.

3. Page 6 : the statement 'Meanwhile, no definitive research exists to elucidate the potential impact of GC content on base editing within the context of GC-rich genomes (> 70%) of *Streptomyces*' may appear abrupt and lacks a causal relationship with the preceding content. To address this gap, it is important to provide additional information regarding the influence of 'GC content' on these high-fidelity variants of SpCas9. Is it solely determined through trial and error?"

4. Page 7 : "CBE3 were codon optimized for *Streptomyces* to enhance their expression in genomes with high GC content, yielding SCBE3 (Fig. 1b)." The CBE3 mentioned here is originally from *Streptomyces* with high GC content, and you should describe the reason for codon optimization in detail.

Minor comments:

1. Page 32 : The formatting of some sentences needs to be adjusted, otherwise it will affect the reading experience, such as the sentence "Streptomyces spores were washed twice in TES (2-[2-

hydroxy-1,1-bis(hydroxymethyl)ethyl)amino]ethanesulfonic acid) buffer (0.05 M, pH 8.0),suspended in 5 mL TES buffer and incubated at 50°C for 10 min to activategermination”.

2. Page 32, : The unit form of numbers needs to be unified and standardized, for example, “25 µg/ml” should be changed to “25 µg/mL”

Point-by-point responses to the reviewers' comments

Reviewer #1:

Wang et al. developed an engineered cytosine base editor (CBE) eSCBE3-NG-Hypa in the important industrial streptomycetes via extensive characterization and screening work. eSCBE3-NG-Hypa was established based on SCBE3 (codon-optimized of CBE3 for *Streptomyces*), in which the nCas9 (D10A) combines multiple excellent traits of Cas9 featured by high-fidelity and an expansive recognition scope, and the deaminase rAPOBEC1 was replaced with hAPOBEC3A(Y130F) with high activity in regions marked by GC motifs. Compared with the current base editors in *Streptomyces*, the engineered CBE eSCBE3-NG-Hypa shows a broad editing scope, high-fidelity, and low bystander activity, and does not discriminate against GC motif or sites with high GC content. Particularly, this editor could achieve base editing of DNA with up to 85% GC content. I believe it could widen the potential applications of precision genome editing in *Streptomyces*. This research work belongs to integrated innovation, with rigorous work design, reliable and detailed data, thorough analysis, and high-quality charts. Overall, this is a high-quality research paper suitable for publication in this high-ranked journal. I have some comments as follows.

Q1: Abstract, Line 6: this sentence should be rephrased; in addition, “active histidine”, “inactive tyrosine”, in which “active” and “inactive” should be removed. Similarly at Page 28, “active His to inactive Tyr”. Amino acids cannot be classified as active or inactive.

A1: Thanks a lot for your comments. The sentence at Line 6 in the abstract ‘Here, we developed an optimal base editing tool, namely eSCBE3-NG-Hypa, based on the editing efficiency and fidelity of the base editor in *Streptomyces*’ has been rephrased as ‘Here, we have developed a base editing tool named eSCBE3-NG-Hypa, tailored with both high-efficiency and -fidelity for *Streptomyces*’ (Line 5-7, Page 2). In addition, the mistakes have been corrected in the revised manuscript.

Q2: Page 24, second paragraph, “100% in group Δ pte (1 out of 1)”, why did the author just analyze one exconjugant? or only 1 exconjugant was obtained from conjugation transfer experiment? Please briefly explain.

A2: During obtaining of *S. avermitilis* mutants, we tried to record the proportion of exconjugants exhibiting base editing events, illustrating the simplicity and efficiency of eSCBE3-NG-Hypa in an application scenario. However, unlike wild-type strain, despite a lot of efforts, only one exconjugant was obtained for screening the mutant Δ pte due to numerous random physical and chemical mutagenesis in the industrial strain *S. avermitilis* 3-115, posing significant challenges for its genetic manipulation. Nevertheless, this exconjugant harbors the expected mutation, thereby achieving the primary experimental objective.

Q3: Fig7a legend. “The sanger sequencing chromatograms illustrate the editing events observed in exconjugants containing uncured plasmids and plasmid-cured mutants”,

please clearly indicate in the figure which chromatograms are the editing events from exconjugants containing uncured plasmids, and which chromatograms are from plasmid-cured mutants.

A3: The Figure 7a has been revised as suggested and also shown as following:

Q4: It's interesting to find that either of inactivation of *olm* and *pte* resulted in over 4-fold increase in avermectin B1a production; however, inactivation of both *olm* and *pte* only led to 1.39-fold increase in avermectin B1a titer. The authors performed WGS analysis of the mutants and they found no unexpected stop codon was introduced in the metabolic regions of the genome of $\Delta olm\Delta pte$ mutant. I wonder whether there are amino acid alterations by detected SNVs in the avermectin biosynthetic genes or known regulatory genes involved in avermectin biosynthesis. Please check it.

A4: Indeed, the analysis of SNVs revealed a common SNV in the *ave* gene cluster for both the *S. avermitilis* mutants Δpte and $\Delta olm\Delta pte$, as depicted in Supplementary Fig. 9c. However, this SNV did not result in any amino acid alterations in the mentioned two mutants or reduce the production of avermectins in the Δpte mutant. This suggests that this single SNV is well tolerated by the strains, and other factors likely contribute to the lower increase in avermectin titers observed in the $\Delta olm\Delta pte$ mutant.

To address this concern, we hypothesized that the dual inactivation of two major biosynthetic gene cluster in strain $\Delta olm\Delta pte$, *olm* and *pte*, led to the massive accumulation of pathway intermediates or products that may be inhibitory or toxic to the cell. Especially, all three gene cluster *ave*, *olm* and *pte* encode type 1 PKSs that utilize common building blocks or intermediates such as malonyl-CoA, methylmalonyl-CoA, propionyl-CoA. Although accumulated building blocks contribute to avermectins production in the $\Delta olm\Delta pte$ mutant, the accumulation of propionyl-CoA and methylmalonyl-CoA could potentially induce growth inhibition for the host organism, confining the increase in avermectins production. This hypothesis is supported by a reference: Zhan, C., Lee, N.,

Lan, G. *et al.* Improved polyketide production in *C. glutamicum* by preventing propionate-induced growth inhibition. *Nat. Metab.* **5**, 1127–1140 (2023).

Moreover, we supplemented additional experiment to prove our assumption. In line with the aforementioned reference, our new experiment reveals a delay in the growth of *S. avermitilis* 3-115 and its corresponding mutants when cultured in TSB medium supplemented with propionate compared to the group without propionate (Supplementary Fig. 12). This observation suggests that metabolites derived from propionate, specifically propionyl-CoA and methylmalonyl-CoA, induce growth inhibition in *S. avermitilis* 3-115 and related strains, potentially constraining the production of secondary metabolites of *Streptomyces*.

These additional results and discussions have been added in the revised manuscript (Page 26-27) and in the revised Supplementary Information as Supplementary Fig. 12 (show here as well).

Supplementary Fig. 12 Propionate derivatives, including propionyl-CoA and methylmalonyl-CoA, induce growth inhibition in *S. avermitilis* 3-115 and its corresponding mutants Δolm , Δpte , and $\Delta olm\Delta pte$. Growth curves of various strains in TSB medium are presented by red connecting lines, while those cultured in TSB medium with 1 g/L propionate are indicated by blue connecting lines. Mean and s.d. shown for $n = 3$ biologically independent samples.

Reviewer #2:

The work describes an advancement on the use of cytosine base editors in *Streptomyces*, which has previously proved challenging due to their high GC content. It is undoubtedly an advance on current methods, improving the options for PAM recognition, increasing efficiency and reducing off target activity. They then use their new methodology to introduce point mutations into avermectin producing streptomyces strains, first in an attempt to make analogues and then to increase production. This latter part is not particularly ground breaking, as it is well known that blocking production of one natural product can switch metabolic flux towards another, but is useful. The data is comprehensive, conclusions valid and methods sufficient.

Q1: What would have been useful to include is a more thorough analysis of how well the exemplar experiments worked with previous best methods. A minor comment in the discussion 'As shown in our prior study, the widely employed BE3 induced unintended off-target effects, ultimately leading to the cessation of avermectin production' suggests this has been done but this references an article that has not undergone peer review and it's hard to make a direct comparison.

A1: Thanks a lot for your constructive suggestions. To comprehensively assess the off-target effects induced by editors, whole-genome sequencing (WGS) normally serves as the gold standard. Tong et al. conducted WGS on *S. coelicolor* treated with CBE3 and identified 22 and 12 SNVs in two pure mutants, respectively (Tong, Y. et al., Highly efficient DSB-free base editing for *Streptomyces* with CRISPR-BEST. *Proc. Natl. Acad. Sci. USA* 116, 20366–20375 (2019)). Although eSCBE-NG-Hypa induced an average of 21.7 SNVs across three independent biological samples, it is important to note that our genomic samples were derived from a mixture of over 100 exconjugants, with SNV frequencies ranging from 12% to 89% (Supplementary Data 3). Our approach has contributed to the detection of more potential SNVs, showing a more comprehensive off-target profile induced by BEs. Additionally, the use of different sgRNAs may lead to varying levels of off-target effects, making direct comparisons challenging between statistical results obtained from different WGS datasets. We considered using our unpublished preprint study (Zhong, Z. et al., Base editing in *Streptomyces* with Cas9-deaminase fusions. *BioRxiv* (2019)), which also employed CBE3 in the same bacterial species *S. avermitilis*, as a comparison to mitigate the impact of genome sequence differences. Importantly, we utilized the same sgRNA to guide eSCBE3-NG-Hypa for target base editing without encountering any unexpected SNVs in the *ave* or *olm* gene clusters. In contrast, our previous study experienced unintended off-target effects with CBE3, leading to the unforeseen cessation of avermectin production. While this experiment was attempted in our unpublished study, it did not yield successful results. Consequently, we endeavored to address the underlying scientific challenges and conducted a comparative analysis. Thus, as suggested, we have revised the discussion in the revised manuscript (*Line 3-19, Page 30*), which now reads as follows:

'To comprehensively assess the off-target effects induced by BEs, WGS serves as the gold

standard. Tong et al. conducted WGS on *S. coelicolor* treated with CBE3 and identified 22 and 12 SNVs in two pure mutants, respectively³⁵. Although eSCBE-NG-Hypa in this study induced an average of 21.7 SNVs across three independent biological samples from *S. coelicolor*, it is important to note that our genomic samples were derived from a mixture of over 100 exconjugants, with SNV frequencies ranging from 12% to 89% (Supplementary Data 3). Our approach has contributed to the detection of more potential SNVs, showing a comprehensive off-target profile induced by BEs. When assessing the applicability of a base editing tool, particularly for highly sequence-similar coding regions such as PKS/NRPS, ensuring its safety is crucial. As our unpublished preprint study, the widely used CBE3 system induced unintended off-target effects at the *aveA4* gene, despite the sgRNA being programmed to target the *olmA1* gene in *S. avermitilis*⁷³. This ultimately led to the cessation of avermectins production. However, the eSCBE3-NG-Hypa developed in this study enabled precise base editing using the same sgRNA without any unexpected SNVs in the *ave* or *olm* gene clusters, as confirmed by whole-genome sequencing data for mutants Δolm . (Supplementary Fig. 9c).’

Q2: Similarly in the section ‘DNA and RNA off-target evaluation of eSCBE3-NG-HF1 and eSCBE3-NG-Hypa’ the control using scBE3 was only discussed for the transcriptomics assessment, not the initial experiment monitoring on-target efficiency. Without these analyses it’s hard to judge whether the improved methodologies truly increase the applicability to an extent sufficient to warrant publication in nature communications.

A2: We agree with your comments and have conducted additional WGS analysis for the SCBE3 experiment group during the revision. As shown in Fig. 4c in the revised manuscript (also see the revised Fig. 4c as following), SCBE3 induced 13.7 ± 0.9 (mean \pm s.e.m.) C-to-T / G-to-A conversions, whereas eSCBE3-NG generated a significantly increased number, with 76.7 ± 9 (mean \pm s.e.m.). This is ascribed to the higher catalytic activity of hAPOBEC3A(Y130F) and the PAM-relaxed capacity of eSCBE3-NG. However, eSCBE3-NG-HF1 and eSCBE3-NG-Hypa, which combine the functionalities of high-fidelity SpCas9-HF1 or HypaCas9, reduced this number to 49.3 ± 14.4 (mean \pm s.e.m.) and 26.3 ± 4.8 (mean \pm s.e.m.), respectively. Based on the supplementary results, we have refined the context in *Page 15*.

Regarding the total number of all types of DNA SNVs, eSCBE3-NG-Hypa did not induce significantly more SNVs than SCBE3, but exhibited the lowest number compared to eSCBE3-NG and eSCBE3-NG-HF1 (see revised Fig. 4d as following and the context in *Page 17*).

According to our supplementary results, we thus revised the discussion in the revised manuscript (*Page 28-29*), which now reads as follows:

‘Although SCBE3 induced the fewest C-to-T / G-to-A conversions among all experimental groups (Fig. 4c), its lower on-target editing efficiency and limited targeting scope should be taken into account. SCBE3 exhibited a mean C-to-T editing rates of 45.7% at the on-target site PSP_CGG-4, whereas that of eSCBE3-NG-Hypa was 56.3% (Supplementary Data 3). Additionally, we demonstrated that SCBE3 discriminates against target sites characterized by non-NGG PAM, high GC content or GC motif, resulting in significantly fewer editable

sites across the whole genome compared to eSCBE3-NG-Hypa. Considering these factors, along with the concern regarding RNA off-target effects of SCBE3 (Fig. 4e and f), we recommend using eSCBE3-NG-Hypa in most application scenarios to achieve high-efficiency and -fidelity in base editing.'

Fig. 4 c The numbers and frequencies of total DNA off-target C-to-T / G-to-A editing induced by the indicated BEs. Boxplot elements shown are the median (midline), the interquartile range of 25% and 75% percentiles (box boundaries). The whiskers mark the 5th and 95th percentiles. **d** Comparison of the total number of detected DNA SNVs induced by indicated BEs. Data are mean \pm s.d. from three independent experiments. ns (not significant), * $P < 0.05$, ** $P < 0.01$.

Q3: My only other minor complaint is that there are an awful lot of acronyms in the article which makes it quite hard to follow which plasmid is which, especially for an article aimed at a general audience. Figure 1 attempts to do this pictorially but it would be useful to show functionally what all the various mutations actually do. At the moment the reader has to look at the figure, check what the colours are, find the corresponding mutations in the figure legend, then check the manuscript to see what these mean. A more succinct summary would be useful – or better yet, a less complicated naming system.

A3: We appreciate your valuable advice. The naming convention for base editors is designed to provide direct comprehension into their respective functions. For example, the designation SCBE3-HF1 signifies that SCBE3 incorporates the capabilities of SpCas9-HF1. Similarly, SCBE3-NG indicates that SCBE3 possesses the attributes of SpCas9-NG, while SCBE3-NG-HF1 denotes that SCBE3 combines the functionalities of both SpCas9-HF1 and SpCas9-NG. Following your suggestion, we have included an explanation of the naming convention of base editors in the revised manuscript (*Line 14-17, Page 8*). It read as:

'The name of base editors is designed to provide direct comprehension into their respective functions. For instance, SCBE3-NG-HF1 denotes that SCBE3 combines the functionalities of both SpCas9-NG and SpCas9-HF1.'

Reviewer #3:

This manuscript focuses on editing tool development of *Streptomyces*, which is known as one of the most prolific producers of antibiotics. By introducing the advance of cytosine base editors, the authors developed a more efficient base editor, namely eSCBE3-NG-Hypa, targeting the *Streptomyces* genome. The authors addressed the problems encountered by the present base editors in their application to *Streptomyces*, specifically the high GC content of its genome that affects the editing range and accuracy of CBE3.

The noteworthy results are that this study combined different cytidine deaminase and Cas9 mutants to finally generate an excellent cytosine base editor eSCBE3-NG-Hypa. Using this tool, the authors successfully introduce stop codons into multiple secondary metabolite synthesis gene clusters, achieving high production of avermectin in *S. avermitilis*. The article follows a clear logical structure encompassing problem finding and problem solving and its engineering application. Moreover, it effectively addresses limitations in CBE for *Streptomyces* genome editing while improving efficiency and fidelity. Therefore, this work is exciting in the streptomyces community. However, considering all the cytidine deaminase and Cas9 mutants used in this work were previously used ones, I suggested the mechanism behind the excellent performance of eSCBE3-NG-Hypa should be further explored in this work.

Response: We acknowledge your advice and are trying to elucidate the mechanisms underlying the function of eSCBE3-NG-Hypa using techniques such as AlphaFold2 or Cryo-EM, with the aim of developing enhanced performance tools. We believe this endeavor holds promise for an intriguing project and would present it in future publications.

Comments:

Q1: In Section 5 of the Results, while Fig. 4c did not provide a comparison of off-target rates for SCBE3 at the DNA level, the evaluation of SCBE3 was conducted at the RNA level in Fig. 4e. It would be valuable to investigate whether the tool developed in this study also exhibits an increase in off-target rates when utilizing Cas9 mutants to expand targeting range. If indeed the off-target rate decreased as depicted in Fig. 4e, it is important to understand the underlying reason.

A1: We agree with your comments and have conducted additional WGS analysis for the SCBE3 experiment group during the revision. As shown in Fig. 4c in the revised manuscript (also see the revised Fig. 4c as following), SCBE3 induced 13.7 ± 0.9 (mean \pm s.e.m.) C-to-T / G-to-A conversions, whereas eSCBE3-NG generated a significantly increased number, with 76.7 ± 9 (mean \pm s.e.m.). This is ascribed to the higher catalytic activity of hAPOBEC3A(Y130F) and the PAM-relaxed capacity of eSCBE3-NG. However, eSCBE3-

NG-HF1 and eSCBE3-NG-Hypa, which combine the functionalities of high-fidelity SpCas9-HF1 or HypaCas9, reduced this number to 49.3 ± 14.4 (mean \pm s.e.m.) and 26.3 ± 4.8 (mean \pm s.e.m.), respectively. Based on the supplementary results, we have refined the context in *Page 15*.

Regarding the total number of all types of DNA SNVs, eSCBE3-NG-Hypa did not induce significantly more SNVs than SCBE3, but exhibited the lowest number compared to eSCBE3-NG and eSCBE3-NG-HF1 (see revised Fig. 4d as following and the context in *Page 17*).

According to our supplementary results, we thus revised the discussion in the revised manuscript (*Page 28-29*), which now reads as follows:

'Although SCBE3 induced the fewest C-to-T / G-to-A conversions among all experimental groups (Fig. 4c), its lower on-target editing efficiency and limited targeting scope should be taken into account. SCBE3 exhibited a mean C-to-T editing rates of 45.7% at the on-target site PSP_CGG-4, whereas that of eSCBE3-NG-Hypa was 56.3% (Supplementary Data 3). Additionally, we demonstrated that SCBE3 discriminates against target sites characterized by non-NGG PAM, high GC content or GC motif, resulting in significantly fewer editable sites across the whole genome compared to eSCBE3-NG-Hypa. Considering these factors, along with the concern regarding RNA off-target effects of SCBE3 (Fig. 4e and f), we recommend using eSCBE3-NG-Hypa in most application scenarios to achieve high-efficiency and -fidelity in base editing.'

Fig. 4 c The numbers and frequencies of total DNA off-target C-to-T / G-to-A editing induced by the indicated BEs. Boxplot elements shown are the median (midline), the interquartile range of 25% and 75% percentiles (box boundaries). The whiskers mark the 5th and 95th percentiles.

d Comparison of the total number of detected DNA SNVs induced by indicated BEs. Data are mean \pm s.d. from three independent experiments. Data are mean \pm s.d. from three independent experiments. ns (not significant), * $P < 0.05$, ** $P < 0.01$.

Q2: Although the application of eSCBE3-NG-Hypa in metabolic engineering of avermectin production demonstrated its powerful, a systematic analysis of the metabolic networks and resource allocation of different genetic modifications would provide a clearer explanation of the reasons for the distinct increases in Fig. 7b.

A2: We appreciate your thoughtful comments. Additional context has been added in the revised manuscript (Page 26-27) and present here as well:

'The metabolic network flux and proteomics analysis of *S. avermitilis* suggest that the efficient production of avermectin B1a relies on the availability of correct precursors in the right ratios: 1 \times methylbutanoyl-CoA + 5 \times methylmalonyl-CoA + 7 \times malonyl-CoA (Gao, Q. et al., Learn from microbial intelligence for avermectins overproduction. *Curr. Opin. Biotechnol.* 48, 251–257 (2017)). Given that the *ave*, *olm*, and *pte* gene clusters encode type 1 PKSs utilizing common building blocks or intermediates like malonyl-CoA, methylmalonyl-CoA, and propionyl-CoA, the disruption of either the *olm* or *pte* competitor pathway leads to the accumulation of avermectin precursors, thereby enhancing avermectin production.'

Q3: Given the unexpected impact of $\Delta olm\Delta pte$ double mutant strain on avermectin titer and production rate in Figure 7b, I suspect there may be some unknown influence by eSCBE3-NG-Hypa.

A3: We have double-checked the SNVs information of the mutant $\Delta olm\Delta pte$, the analysis of SNVs revealed a common SNV in the *ave* gene cluster for both the *S. avermitilis* mutants Δpte and $\Delta olm\Delta pte$, as depicted in Supplementary Fig. 9c. However, this SNV did not result in any amino acid alterations in the mentioned two mutants or reduce the production of avermectins in the Δpte mutant. This suggests that this single SNV is well tolerated by the strains, and other unexpected factors likely contribute to the lower increase in avermectin titers observed in the $\Delta olm\Delta pte$ mutant.

To address this concern, we hypothesized that the dual inactivation of two major biosynthetic gene cluster in strain $\Delta olm\Delta pte$, *olm* and *pte*, led to the massive accumulation of pathway intermediates or products that may be inhibitory or toxic to the cell. Especially, all three gene cluster *ave*, *olm* and *pte* encode type 1 PKSs that utilize common building blocks or intermediates such as malonyl-CoA, methylmalonyl-CoA, propionyl-CoA. Although accumulated building blocks contribute to avermectins production in the $\Delta olm\Delta pte$ mutant, the accumulation of propionyl-CoA and methylmalonyl-CoA could potentially induce growth inhibition for the host organism, confining the increase in avermectins production. This hypothesis is supported by an additional reference: Zhan, C. et al., Improved polyketide production in *C. glutamicum* by preventing propionate-induced growth inhibition. *Nat. Metab.* 5, 1127–1140 (2023).

Moreover, we supplemented additional experiment to prove our assumption. In line

with the aforementioned reference, our new experiment reveals a delay in the growth of *S. avermitilis* 3-115 and its corresponding mutants when cultured in TSB medium supplemented with propionate compared to the group without propionate (Supplementary Fig. 12). This observation suggests that metabolites derived from propionate, specifically propionyl-CoA and methylmalonyl-CoA, induce growth inhibition in *S. avermitilis* 3-115 and related strains, potentially constraining the production of secondary metabolites of *Streptomyces*.

These additional results and discussions have been added in the revised manuscript (Page 26-27) and in the revised Supplementary Information as Supplementary Fig. 12 (show here as well).

Supplementary Fig. 12 Propionate derivatives, including propionyl-CoA and methylmalonyl-CoA, induce growth inhibition in *S. avermitilis* 3-115 and its corresponding mutants Δolm , Δpte , and $\Delta olm\Delta pte$. Growth curves of various strains in TSB medium are presented by red connecting lines, while those cultured in TSB medium with 1 g/L propionate are indicated by blue connecting lines. Mean and s.d. shown for $n = 3$ biologically independent samples.

Q4: Page 6: the statement 'Meanwhile, no definitive research exists to elucidate the potential impact of GC content on base editing within the context of GC-rich genomes (> 70%) of *Streptomyces*' may appear abrupt and lacks a causal relationship with the preceding content. To address this gap, it is important to provide additional information

regarding the influence of 'GC content' on these high-fidelity variants of SpCas9. Is it solely determined through trial and error?"

A4: We are sorry for the ambiguity induced by our imprecise description. In fact, several reported factors affect the editing efficiency of CBE3, including various deaminases, CRISPR proteins, PAM sequences, editing windows, DNA motifs, and more. However, whether the high GC content (> 70%), a hallmark feature of *Streptomyces* genomes, affects the efficiency of base editing remains to be further investigated. We rephrased the statement as follows:

'Moreover, the efficiency of BEs is typically affected by various factors, including different deaminases, CRISPR proteins, PAM constraints, editing window-imposed restrictions, DNA motifs, and more¹⁵. In the case of *Streptomyces*, with a hallmark feature of GC-rich genomes (> 70%), DNA motifs may present a bottleneck for the application of BE3. Widely used rAPOBEC1 of BE3 imposes TC motif preferences while exhibiting poor processing of cytosines within GC motifs^{13,19}, affecting the application of BE3 in *Streptomyces*. Meanwhile, whether the high GC content of protospacers will affect the efficiency of base editing warrants further investigation.' (Page 6)

Q5: Page 7: "CBE3 were codon optimized for *Streptomyces* to enhance their expression in genomes with high GC content, yielding SCBE3 (Fig. 1b)." The CBE3 mentioned here is originally from *Streptomyces* with high GC content, and you should describe the reason for codon optimization in detail.

A5: We have incorporated additional description to clarify the reason for the codon optimization of CBE3 in the revised manuscript (Line 1, Page 8) and also as follows:

'CBE3 was originally developed for use in eukaryotic systems¹³. However, to ensure optimal gene expression and compatibility with high GC content *Streptomyces*, CBE3 was subjected to codon optimization for this bacterial species. This optimization process resulted in the creation of *Streptomyces* CBE3 (SCBE3).'

Minor comments:

Q6: Page 32: The formatting of some sentences needs to be adjusted, otherwise it will affect the reading experience, such as the sentence "*Streptomyces* spores were washed twice in TES (2-[2-hydroxy-1,1-bis(hydroxymethyl)ethyl]amino]ethanesulfonic acid) buffer (0.05 M, pH 8.0), suspended in 5 mL TES buffer and incubated at 50°C for 10 min to activate germination".

A6: Given that TES is a common reagent, we delete its full name in the revised manuscript. Here is the rephrased sentence:

'*Streptomyces* spores were washed twice in TES buffer (0.05 M, pH 8.0), followed by resuspension in 5 mL TES buffer and subsequent incubation at 50 °C for 10 min to induce germination.'

Q7: Page 32: The unit form of numbers needs to be unified and standardized, for example,

“25 µg/ml” should be changed to “25 µg/mL”

A7: We have revised as suggested and double-checked the formatting of the whole manuscript.

REVIEWERS' COMMENTS

Reviewer #1 (Remarks to the Author):

The authors have addressed all the comments I raised previously. Now, I have no other comment. Thank!

Reviewer #2 (Remarks to the Author):

I am generally happy with the modifications to the manuscript.

I think the authors might have misunderstood my point about the naming of the editors. I understand how they have derived the names, and don't think their additional explanatory sentence is required. I mean to say that it would be useful to have a figure that explains what each part of the name means, so it can be easily referred to. At the moment if I am half way through reading it and read SCBE3-NG-HF1 I have to remember what SCBE3 is, what NG is and what HF1 is, which are all in different locations in the text. For someone who works in this area it might be obvious, but for a broad audience it will make the manuscript very hard to comprehend.

Reviewer #3 (Remarks to the Author):

The authors appear to have tackled all the concerns I have, as well as those raised by other reviewers.

Point-by-point responses to the reviewers' comments

Reviewer #1:

The authors have addressed all the comments I raised previously. Now, I have no other comment. Thank!

Response: We would like to thank the reviewer for constructive feedback throughout the review process, that has improved our manuscript.

Reviewer #2:

I am generally happy with the modifications to the manuscript.

I think the authors might have misunderstood my point about the naming of the editors. I understand how they have derived the names, and don't think their additional explanatory sentence is required. I mean to say that it would be useful to have a figure that explains what each part of the name means, so it can be easily referred to. At the moment if I am half way through reading it and read SCBE3-NG-HF1 I have to remember what SCBE3 is, what NG is and what HF1 is, which are all in different locations in the text. For someone who works in this area it might be obvious, but for a broad audience it will make the manuscript very hard to comprehend.

Response: We appreciate the reviewer for the constructive suggestions. We have deleted additional explanatory sentence for the naming of the base editors and revised Figure 1b as following for better understanding.

Reviewer #3:

The authors appear to have tackled all the concerns I have, as well as those raised by other reviewers.

Response: We thank the reviewer for the positive comment and for all the suggestions in the review process that helped us to improve our manuscript.